# Noise Conditional Variational Score Distillation

**Xinyu Peng** [1]  **Ziyang Zheng** [2]  **Yaoming Wang** [3]  **Han Li** [2]  **Nuowen Kan** [2]  **Wenrui Dai** [1]
**Chenglin Li** [2]  **Junni Zou** [1]  **Hongkai Xiong** [2]

{`xypeng9903, zhengziyang, qingshi9974, kannw_1230, daiwenrui,`
`lcl1985, zoujunni, xionghongkai`}`@sjtu.edu.cn; wangyaoming03@meituan.com`

## Abstract

We propose Noise Conditional Variational Score Distillation (NCVSD), a novel method for distilling pretrained diffusion models into generative denoisers. We achieve this by revealing that the unconditional score function implicitly characterizes the score function of denoising posterior distributions. By integrating this insight into the Variational Score Distillation (VSD) framework, we enable scalable learning of generative denoisers capable of approximating samples from the denoising posterior distribution across a wide range of noise levels. The proposed generative denoisers exhibit desirable properties that allow fast generation while preserve the benefit of iterative refinement: (1) fast one-step generation through sampling from pure Gaussian noise at high noise levels; (2) improved sample quality by scaling the test-time compute with multi-step sampling; and (3) zero-shot probabilistic inference for flexible and controllable sampling. We evaluate NCVSD through extensive experiments, including class-conditional image generation and inverse problem solving. By scaling the test-time compute, our method outperforms teacher diffusion models and is on par with consistency models of larger sizes. Additionally, with significantly fewer NFEs than diffusion-based methods, we achieve record-breaking LPIPS on inverse problems. The source code is available at https://github.com/xypeng9903/ncvsd.

[1]Department of Computer Science and Engineering, Shanghai Jiao Tong University, Shanghai, China [2]Department of Electronic Engineering, Shanghai Jiao Tong University, Shanghai, China [3]Meituan Inc, China. Correspondence to: Ziyang Zheng <zhengziyang@sjtu.edu.cn>, Yaoming Wang <wangyaoming03@meituan.com>, Wenrui Dai <daiwenrui@sjtu.edu.cn>.

*Proceedings of the 42nd International Conference on Machine Learning*, Vancouver, Canada. PMLR 267, 2025. Copyright 2025 by the author(s).

## 1. Introduction

Diffusion models (Song & Ermon, 2019; Ho et al., 2020; Song et al., 2021b; Karras et al., 2022), also known as score-based generative models, have emerged as a dominant paradigm for high-dimensional data generation. A defining characteristic lies in their inherently iterative sampling mechanism, offering unprecedented flexibility for inference-time control. This characteristic facilitates several advantages, including the flexible trade-off between computational complexity and sample quality, as well as enabling zero-shot controllable sampling across a wide range of downstream tasks (Chung et al., 2023; Yu et al., 2023; Song et al., 2023b; Uehara et al., 2025). Nevertheless, the iterative process suffers from significant limitations regarding sampling efficiency and real-time applications. To address this issue, recent years have witnessed a surge of interest in distilling teacher diffusion models into fast generators (Luo et al., 2023; Yin et al., 2024b; Zhou et al., 2024; Sauer et al., 2024). However, these generators support only one or a few sampling steps, thereby discarding the capacity for iterative refinement of generated samples, which is especially crucial for imperfectly trained generators and zero-shot controllable sampling. This raises a fundamental question: *Can we develop a generative model that enables fast generation while preserving the desirable attributes of iterative refinement?*

To tackle this challenge, we develop *generative denoisers* — a class of models specifically designed to generate approximate samples from the denoising posterior distributions. Well-trained generative denoisers demonstrate a unique combination of advantages that effectively balance sampling efficiency and flexibility. Specifically, **i)** they enable efficient one-step unconditional generation by sampling from pure Gaussian noise; **ii)** they inherently support multi-step sampling with improved sample quality, offering a paradigm shift from traditional training-time computation to flexible test-time computation (OpenAI, 2024; Geng et al., 2025); **iii)** they facilitate *asymptotic exact* probabilistic inference through seamless integration with the *Split Gibbs Sampler* framework (Vono et al., 2019), as illustrated in Figure 1.

To train generative denoisers, we first establish a funda-

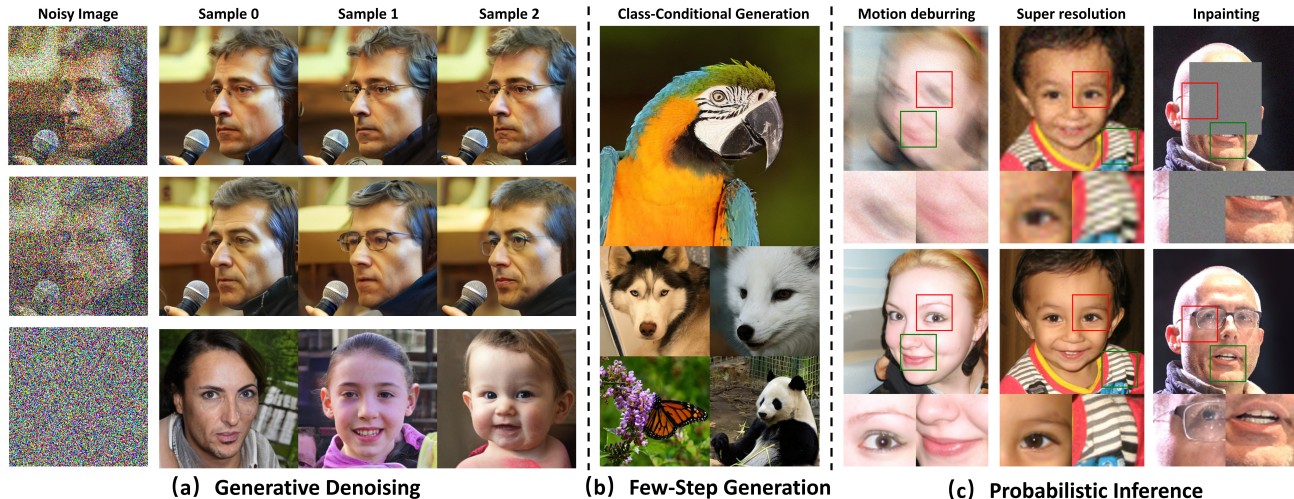

| Noisy Image | Sample 0 | Sample 1 | Sample 2 | Class-Conditional Generation | Motion deburring | Super resolution | Inpainting |

**(a)** Generative Denoising     **(b)** Few-Step Generation     **(c)** Probabilistic Inference

*Figure 1.* The proposed generative denoisers, distilled from pretrained diffusion models, support a variety of tasks. In (a), the generative denoiser demonstrates the ability to generate diverse samples that approximate the denoising posterior distribution at arbitrary noise levels. In (b), we present the 4-step class-conditional generation results on the ImageNet-512×512 dataset. In (c), we showcase the plug-and-play probabilistic inference capability of the generative denoiser.

mental theoretical connection by demonstrating that the unconditional score function inherently characterizes the score function of denoising posterior distributions. This insight enables us to extend conventional Variational Score Distillation (VSD) (Wang et al., 2024) by explicitly conditioning on noisy data, emerging as a novel diffusion distillation approach, namely Noise Conditional VSD (NCVSD). Furthermore, we introduce an auxiliary adversarial loss to facilitate learning from real data, thereby overcoming the performance upper bound imposed by the teacher diffusion models. Finally, we meticulously engineer the parameterization of generative denoisers, which not only enables efficient knowledge transfer from the teacher diffusion model but also leverages the inductive bias of preconditioning, as elucidated by Karras et al. (2022).

To demonstrate the effectiveness of NCVSD, we evaluate the distilled generative denoisers via extensive experiments, including class-conditional image generation on ImageNet-64×64 and ImageNet-512×512 datasets, as well as solving a wide range of linear and nonlinear inverse problems. In the class-conditional image generation task, we observed that generative denoisers beat consistency models (CM) (Song et al., 2023c) with state-of-the-art training method sCM (Lu & Song, 2025) of similar sizes. In addition, we observed that by scaling the test-time compute, generative denoisers can achieve performance comparable to sCM of larger sizes. For example, on ImageNet-512×512 dataset, the 4-step FID of generative denoiser (1.73), distilled from EDM2-L (Karras et al., 2024), surpasses the FID of sCM (1.88) distilled from EDM2-XXL. In inverse problem solving tasks, we observed that PnP-GD, the proposed plug-and-play method

for solving inverse problems using our generative denoisers, achieves competitive results compared to state-of-the-art diffusion-based inverse problem solvers (Chung et al., 2023; Wu et al., 2024; Zhang et al., 2024), while reducing the required number of function evaluations (NFE) by an order of magnitude. In particular, we achieve record-breaking LPIPS performance on a range of linear inverse problems. For challenging nonlinear inverse problems, current diffusion-based solvers typically require 1k NFE to produce reasonable results, whereas the proposed method only requires 50 NFE.

## 2. Backgrounds

### 2.1. Diffusion Models

Diffusion models aim to generate samples that approximate the target data distribution $q_{\text{data}}(\mathbf{x}_0)$. To achieve this, they employ a family of Gaussian perturbation kernels defined as $q(\mathbf{x}_t|\mathbf{x}_0) = \mathcal{N}(\mathbf{x}_t|\mathbf{x}_0, \sigma_t^2\mathbf{I})$, which gradually perturb the original distribution into a sequence of noisy distributions $q(\mathbf{x}_t) = \mathbb{E}_{q_{\text{data}}(\mathbf{x}_0)}[q(\mathbf{x}_t|\mathbf{x}_0)]$. The noise scale $\sigma_t$ is designed to monotonically increase with the diffusion time step $t$. Through this progressive perturbation process, the final distribution $q(\mathbf{x}_T)$ at terminal time $T$ becomes sufficiently close to the isotropic Gaussian distribution $\mathcal{N}(\mathbf{0}, \sigma_T^2\mathbf{I})$. In this paper, we adopt $\sigma_t = t$ following the approach in (Karras et al., 2022) for simplicity.

The main idea of diffusion models is to find a way to sample from $q(\mathbf{x}_t)$ with annealing decreasing noise levels, such that at the end of the sampling process the distribution of the samples, $q(\mathbf{x}_{t_{\min}})$, will be close to the original data distribution $q_{\text{data}}(\mathbf{x}_0)$. One notable example is the *Probability*

*Flow Ordinary Differential Equation* (PF-ODE) (Song et al., 2021b; Karras et al., 2022), which takes the following form:

$$d\mathbf{x}_t = -t\nabla_{\mathbf{x}_t} \log q(\mathbf{x}_t)dt, \ \ \mathbf{x}_T \sim q(\mathbf{x}_T), \qquad (1)$$

where the unknown gradient of the log density $\nabla \log q(\mathbf{x}_t)$, also known as the score function, can be linked to the conditional expectation through Tweedie's formula (Efron, 2011):

$$\nabla_{\mathbf{x}_t} \log q(\mathbf{x}_t) = t^{-2}\left(\mathbb{E}[\mathbf{x}_0|\mathbf{x}_t] - \mathbf{x}_t\right). \qquad (2)$$

Therefore, the score function can be estimated by training a neural network $D_\phi(\mathbf{x}_t, t) \approx \mathbb{E}[\mathbf{x}_0|\mathbf{x}_t]$ with input $\mathbf{x}_t$ and $t$ and parameters $\phi$, referred to as the score model, through a simple regression objective as $\min_\phi \mathbb{E}_{t,q_{\text{data}}(\mathbf{x}_0)q(\mathbf{x}_t|\mathbf{x}_0)}\left[\|\mathbf{x}_0 - D_\phi(\mathbf{x}_t, t)\|_2^2\right].$

## 2.2. Variational Score Distillation

To address the slow inference speed of diffusion models, recent studies have explored methods for distilling diffusion models into GAN-like generators $\mathbf{x}_0 = G_\theta(\mathbf{z})$, $\mathbf{z} \sim \mathcal{N}(\mathbf{0}, \mathbf{I})$. Notably, Variational Score Distillation (VSD), initially developed for 3D generation in Prolific-Dreamer (Wang et al., 2024), has been successfully adapted to accelerate image generation through diffusion model distillation (Luo et al., 2023; Yin et al., 2024b;a; Nguyen & Tran, 2024). The objective of VSD is to minimize the reversed KL divergence of the diffused model distribution and the diffused data distribution[1]:

$$\min_\theta \mathcal{L}_{\text{vsd}}(\theta) := \mathbb{E}_t[D_{KL}(p_\theta(\mathbf{x}_t)||q(\mathbf{x}_t))], \qquad (3)$$

where $p_\theta(\mathbf{x}_t) := \mathbb{E}_{\mathbf{z},\mathbf{x}_0=G_\theta(\mathbf{z})}[q(\mathbf{x}_t|\mathbf{x}_0)]$ and $q(\mathbf{x}_t) := \mathbb{E}_{q_{\text{data}}(\mathbf{x}_0)}[q(\mathbf{x}_t|\mathbf{x}_0)]$ are defined by adding Gaussian noise $\mathcal{N}(0, t^2\mathbf{I})$ on the generated data $\mathbf{x}_0 = G_\theta(\mathbf{z})$ and real data $\mathbf{x}_0 \sim p_{\text{data}}$, respectively. The fundamental result of VSD is that the gradient of the VSD objective can be linked to the score functions (Wang et al., 2024; Luo et al., 2023):

$$\nabla_\theta \mathcal{L}_{\text{vsd}}(\theta) = \mathbb{E}_{t,\mathbf{z},\mathbf{x}_0=G_\theta(\mathbf{z}),\mathbf{x}_t\sim q(\mathbf{x}_t|\mathbf{x}_0)}$$
$$\left[(\nabla_{\mathbf{x}_t} \log p_\theta(\mathbf{x}_t) - \nabla_{\mathbf{x}_t} \log q(\mathbf{x}_t))\frac{\partial G_\theta(\mathbf{z})}{\partial \theta}\right], \quad (4)$$

where $\nabla_{\mathbf{x}_t} \log q(\mathbf{x}_t)$ can be estimated using a pretrained score model, and $\nabla_{\mathbf{x}_t} \log p_\theta(\mathbf{x}_t)$ can be estimated via an auxiliary score model for the generated data $\mathbf{x}_0 = G_\theta(\mathbf{z})$, with the training of the score model being conducted online with the training of the generator $G_\theta(\mathbf{z})$.

## 2.3. Diffusion-based Posterior Sampling

In practical applications, sampling from a posterior distribution $q(\mathbf{x}_0|\mathbf{y}) \propto q_{\text{data}}(\mathbf{x}_0)q(\mathbf{y}|\mathbf{x}_0)$ given conditions $\mathbf{y}$ in of

---

[1]Without loss of generality, we omit the weighting functions for KL divergences across different $t$, as they can be transformed into the density of $t$ or considered as the importance weight.

great interest. Diffusion methods have recently been widely adopted for posterior sampling, but existing approaches often involve trade-offs between flexibility, posterior exactness, and computational efficiency. Supervised methods (Saharia et al., 2022; Rombach et al., 2022) lack flexibility, as they require retraining for each specific task. Zero-shot methods offer greater flexibility by approximating the conditional score from a pretrained unconditional one, but they introduce irreducible errors by approximating the denoising posterior with Dirac (Chung et al., 2023) or Gaussian distributions (Song et al., 2023b; Peng et al., 2024). Recently, asymptotically exact methods, such as PnP-DM (Wu et al., 2024), ensure exact posterior sampling in the asymptotic limit but are computationally expensive, relying on reverse diffusion simulations for sampling from denoising posterior distributions that requires a large number of NFEs.

In this paper, we propose a posterior sampling method that strikes a balance between flexibility, posterior exactness, and computational efficiency. To achieve this, we first establish a fundamental theoretical connection, demonstrating that the conditional score function conditioned on noisy data has a tractable closed-form solution that can be precisely expressed using the unconditional score function. Building on this result, we distill a one-step generative denoiser capable of efficiently sampling from the denoising posterior distribution across a wide range of noise levels. Leveraging this efficient generative denoiser, we address the slow reverse diffusion simulations in PnP-DM through a simple plug-in replacement of the generative denoiser. This results in an efficient and flexible posterior sampling method with asymptotically exact guarantees.

## 3. Noise Conditional Variational Score Distillation

### 3.1. Conditioning VSD on Noisy Data

The core objective of NCVSD is to learn a conditional generative model $\mu_\theta(\mathbf{x}_0|\mathbf{y}_\sigma)$ that generates samples approximating the denoising posterior distribution $q(\mathbf{x}_0|\mathbf{y}_\sigma) \propto q_{\text{data}}(\mathbf{x}_0)\mathcal{N}(\mathbf{y}_\sigma|\mathbf{x}_0, \sigma^2\mathbf{I})$ across a wide range of noise levels $\sigma > 0$, where $\mathbf{y}_\sigma$ denotes the noisy data generated by adding Gaussian noise on $\mathbf{x}_0$. To this end, we propose to solve the following reverse KL minimization problem:

$$\min_\theta \mathbb{E}_\sigma \mathbb{E}_{q(\mathbf{y}_\sigma)}\left[D_{KL}(\mu_\theta(\mathbf{x}_0|\mathbf{y}_\sigma)||q(\mathbf{x}_0|\mathbf{y}_\sigma))\right], \quad (5)$$

where $\sigma$ is drawn from a predefined distribution of noise levels, $q(\mathbf{y}_\sigma) = \mathbb{E}_{q_{\text{data}}(\mathbf{x}_0)}[\mathcal{N}(\mathbf{y}_\sigma|\mathbf{x}_0, \sigma^2\mathbf{I})]$ is the marginal distribution of $\mathbf{y}_\sigma$, and $\mu_\theta(\mathbf{x}_0|\mathbf{y}_\sigma)$ is defined using implicit distribution induced from a conditional one-step generator as $\mathbf{x}_0 = G_\theta(\mathbf{y}_\sigma, \sigma, \mathbf{z}), \mathbf{z} \sim \mathcal{N}(\mathbf{0}, \mathbf{I})$.

However, directly optimizing Equation (5) is challenging, as the high-density regions of $q(\mathbf{x}_0|\mathbf{y}_\sigma)$ may be extremely

sparse in high-dimensional space (Song & Ermon, 2019; Wang et al., 2024). Inspired by VSD, we diffuse the original distributions $\mu_\theta(\mathbf{x}_0|\mathbf{y}_\sigma)$ and $q(\mathbf{x}_0|\mathbf{y}_\sigma)$ using Gaussian kernels $q(\mathbf{x}_t|\mathbf{x}_0) = \mathcal{N}(\mathbf{x}_t|\mathbf{x}_0, t^2\mathbf{I})$, to construct an alternative optimization problem. Specifically, we define

$$p_\theta(\mathbf{x}_t|\mathbf{y}_\sigma) := \mathbb{E}_{\mu_\theta(\mathbf{x}_0|\mathbf{y}_\sigma)}\left[q(\mathbf{x}_t|\mathbf{x}_0)\right], \qquad (6)$$

$$q(\mathbf{x}_t|\mathbf{y}_\sigma) := \mathbb{E}_{q(\mathbf{x}_0|\mathbf{y}_\sigma)}\left[q(\mathbf{x}_t|\mathbf{x}_0)\right]. \qquad (7)$$

We then minimize the reverse KL divergence between $p_\theta(\mathbf{x}_t|\mathbf{y}_\sigma)$ and $q(\mathbf{x}_t|\mathbf{y}_\sigma)$ for all $t$:

$$\min_\theta \mathcal{L}_{\text{ncvsd}}(\theta) := \mathbb{E}_{t,\sigma,\mathbf{y}_\sigma}[D_{KL}(p_\theta(\mathbf{x}_t|\mathbf{y}_\sigma)||q(\mathbf{x}_t|\mathbf{y}_\sigma)]. \ (8)$$

Similar to Equation (4) for VSD, the gradient of the NCVSD loss relates to *conditional* score functions:

$$\nabla_\theta \mathcal{L}_{\text{ncvsd}}(\theta) = \mathbb{E}_{t,\sigma,\mathbf{y}_\sigma,\mathbf{z},\mathbf{x}_0=G_\theta(\mathbf{y}_\sigma,\sigma,\mathbf{z}),\mathbf{x}_t\sim q(\mathbf{x}_t|\mathbf{x}_0)}$$
$$\left[(\nabla_{\mathbf{x}_t}\log p_\theta(\mathbf{x}_t|\mathbf{y}_\sigma) - \nabla_{\mathbf{x}_t}\log q(\mathbf{x}_t|\mathbf{y}_\sigma))\frac{\partial G_\theta(\mathbf{y}_\sigma,\sigma,\mathbf{z})}{\partial\theta}\right]. \ (9)$$

We denote $\nabla_{\mathbf{x}_t}\log p_\theta(\mathbf{x}_t|\mathbf{y}_\sigma)$ and $\nabla_{\mathbf{x}_t}\log q(\mathbf{x}_t|\mathbf{y}_\sigma)$ as the *model score* and *data score*, where the former represents the score function of the generative denoiser's output distribution and the latter corresponds to the data distribution. The derivation of Equation (9) is provided in Appendix A.1.

**Estimate Data Score:** The pretrained score model, which solely estimates $\nabla_{\mathbf{x}_t}\log q(\mathbf{x}_t)$, cannot be direclty utilized to predict $\nabla_{\mathbf{x}_t}\log q(\mathbf{x}_t|\mathbf{y}_\sigma)$. We address this by showing that $\nabla_{\mathbf{x}_t}\log q(\mathbf{x}_t|\mathbf{y}_\sigma)$ has a tractable closed-form solution, which can be exactly represented by $\nabla_{\mathbf{x}_t}\log q(\mathbf{x}_t)$, as demonstrated in Proposition 1.

**Proposition 1.** *Suppose $(\mathbf{x}_0, \mathbf{y}_\sigma, \mathbf{x}_t)$ follow the joint distribution $q_{data}(\mathbf{x}_0)\mathcal{N}(\mathbf{y}_\sigma|\mathbf{x}_0, \sigma^2\mathbf{I})\mathcal{N}(\mathbf{x}_t|\mathbf{x}_0, t^2\mathbf{I})$. For any $\rho > 0$, define the denoising posterior of $\mathbf{x}_0$ with noise level $\rho$ as $q(\mathbf{x}_0|\mathbf{y}_\rho) \propto q_{data}(\mathbf{x}_0)\mathcal{N}(\mathbf{y}_\rho|\mathbf{x}_0, \rho^2\mathbf{I})$. We obtain that*

$$q(\mathbf{x}_0|\mathbf{x}_t, \mathbf{y}_\sigma) = q\left(\mathbf{x}_0|\mathbf{y}_{\sigma_{\text{eff}}}\right), \qquad (10)$$

$$\nabla_{\mathbf{x}_t}\log q(\mathbf{x}_t|\mathbf{y}_\sigma) = t^{-2}\left(\mathbb{E}\left[\mathbf{x}_0|\mathbf{y}_{\sigma_{\text{eff}}}\right] - \mathbf{x}_t\right), \qquad (11)$$

*where $\mathbf{y}_{\sigma_{\text{eff}}} = \frac{\sigma^{-2}\mathbf{y}_\sigma + t^{-2}\mathbf{x}_t}{\sigma^{-2}+t^{-2}}$, and $\sigma_{\text{eff}} = (\sigma^{-2} + t^{-2})^{-\frac{1}{2}}$ is the noise level of $q(\mathbf{x}_0|\mathbf{y}_{\sigma_{\text{eff}}})$, which is referred to as the effective noise level.*

Please refer to Appendix A.2 for the proof. Note that the unconditonal score function can be linked with $\mathbb{E}[\mathbf{x}_0|\mathbf{y}_{\sigma_{\text{eff}}}]$ according to Equation (2). By Proposition 1, estimating $\nabla_{\mathbf{x}_t}\log q(\mathbf{x}_t|\mathbf{y}_\sigma)$ can be reduced to leveraging a pretrained unconditional score model $D_0(\mathbf{y}_\rho, \rho) \approx \mathbb{E}[\mathbf{x}_0|\mathbf{y}_\rho]$, which is readily available in many settings. This leads to our proposed estimator for $\nabla\log q(\mathbf{x}_t|\mathbf{y}_\sigma)$ using $D_0$:

$$t^{-2}\left(D_0\left(\tfrac{\sigma^{-2}\mathbf{y}_\sigma + t^{-2}\mathbf{x}_t}{\sigma^{-2}+t^{-2}}, (\sigma^{-2} + t^{-2})^{-\frac{1}{2}}\right) - \mathbf{x}_t\right). \quad (12)$$

**Estimate Model Score:** $\nabla\log p_\theta(\mathbf{x}_t|\mathbf{y}_\sigma)$ can be estimated by training a *conditional* score model $D_\phi(\mathbf{x}_t, t, \mathbf{y}_\sigma, \sigma)$ using common diffusion objective:

$$\min_\phi \mathbb{E}_{\mu_\theta(\mathbf{x}_0|\mathbf{y}_\sigma)q(\mathbf{x}_t|\mathbf{x}_0)}\left[\|\mathbf{x}_0 - D_\phi(\mathbf{x}_t, t, \mathbf{y}_\sigma, \sigma)\|_2^2\right]. \ (13)$$

It is worth mentioning that $\nabla\log p_\theta(\mathbf{x}_t|\mathbf{y}_\sigma)$ cannot be estimated in the same manner as $\nabla_{\mathbf{x}_t}\log q(\mathbf{x}_t|\mathbf{y}_\sigma)$. The joint distribution of $(\mathbf{y}_\sigma, \mathbf{x}_0)$ is $q(\mathbf{y}_\sigma)\mu_\theta(\mathbf{x}_0|\mathbf{y}_\sigma)$, which differs from the assumption in Proposition 1. As a result, the distribution of $\mathbf{y}_\sigma$ given $\mathbf{x}_0$ is no longer Gaussian.

### 3.2. Auxiliary Adversarial Loss

The effectiveness of the NCVSD gradient in Equation (9) critically depends on accurate estimation of both the data score and the model score. However, in practice, the data score provided by the pre-trained teacher score model is often imperfect, and accurately estimating the model score is challenging due to the evolving model parameter $\theta$ during training. To address these issues, prior works have shown that incorporating an auxiliary adversarial loss can improve performance by leveraging real data in addition to the teacher model (Kim et al., 2024; Yin et al., 2024b; Zhou et al., 2025; Sauer et al., 2024). Inspired from these approaches, we propose minimizing the Jensen-Shannon divergence (JSD) alongside the KL divergence minimization in Equation (8):

$$\min_\theta \mathcal{L}_{\text{adv}}(\theta) := \mathbb{E}_{t,\sigma,\mathbf{y}_\sigma}[D_{JS}(p_\theta(\mathbf{x}_t|\mathbf{y}_\sigma)||q(\mathbf{x}_t|\mathbf{y}_\sigma)]. \ (14)$$

To optimize Equation (14), we introduce a classification neural network, $C_\psi$, referred to as the discriminator, to convert the JSD minimization (Equation (14)) into a tractable adversarial optimization problem (Goodfellow et al., 2014):

$$\min_\theta \max_\psi \mathbb{E}_{t,\sigma,\mathbf{y}_\sigma}\Big[\mathbb{E}_{q(\mathbf{x}_0|\mathbf{y}_\sigma)q(\mathbf{x}_t|\mathbf{x}_0)}[\log C_\psi(\mathbf{x}_t, t, \mathbf{y}_\sigma, \sigma)]$$
$$+ \mathbb{E}_{\mu_\theta(\mathbf{x}_0|\mathbf{y}_\sigma)q(\mathbf{x}_t|\mathbf{x}_0)}[\log(1 - C_\psi(\mathbf{x}_t, t, \mathbf{y}_\sigma, \sigma))]\Big]. \ (15)$$

Although sampling from $q(\mathbf{x}_0|\mathbf{y}_\sigma)$ is intractable, we can apply the bidirectional Monte Carlo method (Grosse et al., 2016; Zhao et al., 2024) by leveraging the fact that $q(\mathbf{y}_\sigma)q(\mathbf{x}_0|\mathbf{y}_\sigma) = q_{\text{data}}(\mathbf{x}_0)\mathcal{N}(\mathbf{y}_\sigma|\mathbf{x}_0, \sigma^2\mathbf{I})$. This enables us to compute an unbiased estimate of Equation (15) using samples from the training data distribution $q_{\text{data}}(\mathbf{x}_0)$.

Accordingly, we employ a weighted sum of Equations (8) and (15) as the training loss to implement NCVSD. To effectively balance the contributions of $\mathcal{L}_{\text{ncvsd}}$ and $\mathcal{L}_{\text{adv}}$, we employ *uncertainty weighting* (Kendall et al., 2018; Karras et al., 2024; Lu & Song, 2025) to transform the optimization of $\mathcal{L}_{\text{ncvsd}}$ into a form that resembles a Gaussian log-likelihood. Additionally, we scale $\mathcal{L}_{\text{adv}}$ by the number of

the data dimensions, ensuring that both losses are approximately on the same scale. For further details and pseudo code or a single iteration of the NCVSD training iteration, please refer to Appendix B.2 and Algorithm 3.

### 3.3. Multi-Step Sampling

The proposed generative denoiser also support multi-step sampling, allowing a trade-off between sample quality and inference cost. To this end, we introduce latent variables $\mathbf{x}_{1:N}$[2] following the Denoising Diffusion Implicit Model (DDIM) (Song et al., 2021a) to extend $q(\mathbf{x}_0|\mathbf{y}_\sigma)$ into $q(\mathbf{x}_{0:N}|\mathbf{y}_\sigma)$ as $q(\mathbf{x}_{0:N}|\mathbf{y}_\sigma) := q(\mathbf{x}_0|\mathbf{y}_\sigma)q(\mathbf{x}_{1:N}|\mathbf{x}_0)$, and $q(\mathbf{x}_{1:N}|\mathbf{x}_0) := q(\mathbf{x}_N|\mathbf{x}_0)\prod_{i=2}^{N}q(\mathbf{x}_{i-1}|\mathbf{x}_i,\mathbf{x}_0)$. Accordingly, the reverse process is defined as $p(\mathbf{x}_{0:N}) = p(\mathbf{x}_N)\prod_{i=1}^{N}p(\mathbf{x}_{i-1}|\mathbf{x}_i)$. To enable a gradual approximation of $q(\mathbf{x}_0|\mathbf{y}_\sigma)$ using an annealing decreasing noise schedule $\{\sigma_i\}_{i=1}^{N}$ as $i$ decreases during the multi-step sampling, we construct the forward and reverse processes such that the marginal distributions $p(\mathbf{x}_i)$ equal to $q(\mathbf{x}_i|\mathbf{y}_\sigma)$ for all $i$, as formalized in Proposition 2.

**Proposition 2.** *By constructing the following distributions:*

$$q(\mathbf{x}_N|\mathbf{x}_0) = \mathcal{N}(\mathbf{x}_0, \sigma_N^2\mathbf{I}),$$
$$q(\mathbf{x}_{i-1}|\mathbf{x}_i,\mathbf{x}_0) = \mathcal{N}\left(\mathbf{x}_0 + \sigma_{i-1}\sqrt{1-\zeta}\cdot\frac{\mathbf{x}_i-\mathbf{x}_0}{\sigma_i}, \sigma_{i-1}^2\zeta\mathbf{I}\right),$$
$$p(\mathbf{x}_N) = \mathbb{E}_{q(\mathbf{x}_0|\mathbf{y}_\sigma)}[q(\mathbf{x}_N|\mathbf{x}_0)],$$
$$p(\mathbf{x}_{i-1}|\mathbf{x}_i) = \mathbb{E}_{q(\mathbf{x}_0|\mathbf{x}_i,\mathbf{y}_\sigma)}[q(\mathbf{x}_{i-1}|\mathbf{x}_i,\mathbf{x}_0)],$$

*we have that for any $\zeta \in (0,1]$, $q(\mathbf{x}_i|\mathbf{x}_0) = \mathcal{N}(\mathbf{x}_i|\mathbf{x}_0, \sigma_i^2\mathbf{I})$ and $p(\mathbf{x}_i) = q(\mathbf{x}_i|\mathbf{y}_\sigma)$ for $i = 1, 2, ..., N$. In addition, the following equality holds:*

$$q(\mathbf{x}_0|\mathbf{x}_i,\mathbf{y}_\sigma) = q\left(\mathbf{x}_0 \mid \mathbf{y}_{\sigma_{\text{eff}}} = \frac{\sigma^{-2}\mathbf{y}_\sigma + \sigma_i^{-2}\mathbf{x}_i}{\sigma^{-2}+\sigma_i^{-2}}\right), \quad (16)$$

*and the effective noise level $\sigma_{\text{eff}} = (\sigma^{-2}+\sigma_i^{-2})^{-\frac{1}{2}}$.*

Please refer to Appendix A.3 for the proof. According to Proposition 2, multi-step sampling for the generative denoiser can be achieved by sampling recursively from $p(\mathbf{x}_{i-1}|\mathbf{x}_i)$ as iteratively perform the following two steps:

1. sampling $\mathbf{x}_0$ from $\mu_\theta(\mathbf{x}_0|\mathbf{y}_{\sigma_{\text{eff}}})$ according to Equation (16);
2. sampling $\mathbf{x}_{i-1}$ from $q(\mathbf{x}_{i-1}|\mathbf{x}_i,\mathbf{x}_0)$;

The pseudocode for multi-step sampling is presented in Algorithm 2.

---

[2]Note that $\{\mathbf{x}_i\}_{i=0}^{N}$ and $\{\mathbf{x}_t\}_{t\in[0,T]}$ differ. The discrete-time latent variable $\mathbf{x}_i$ is introduced exclusively during the sampling phase, whereas $\mathbf{x}_t$ denotes a continuous-time, noise-corrupted data variable that exists solely during the training process. Besides, the noise level corresponding to $\mathbf{x}_i$ is $\sigma_i$, while that of $\mathbf{x}_t$ is $t$.

---

**Algorithm 1** Probablistic inference with PnP-GD

**Input:** generative denoiser $\mu_\theta(\mathbf{x}_0|\mathbf{y}_\sigma)$, energy function $\frac{1}{\beta}\mathcal{E}(\cdot)$, noise annealing schedule $\sigma_N > ... > \sigma_1 \approx 0$,
$\mathbf{u}^N \sim \mathcal{N}(\mathbf{0}, \sigma_N^2\mathbf{I})$
**for** $i = N, ..., 2$ **do**
  $\mathbf{x}_0^i \sim \mu_\theta(\mathbf{x}_0|\mathbf{y}_{\sigma_i} = \mathbf{u}^i)$ or multi-step sampling
  **if** $\sigma_i < \sigma_{\text{ema}}$ **then**
    $\mathbf{x}_0 \leftarrow \mu \cdot \mathbf{x}_0 + (1-\mu) \cdot \mathbf{x}_0^i$
  **else**
    $\mathbf{x}_0 \leftarrow \mathbf{x}_0^i$
  **end if**
  $\mathbf{u}^{i-1} \sim \exp\left(-\frac{1}{\beta}\mathcal{E}(\mathbf{u}^{i-1}) - \frac{1}{2\sigma_{i-1}^2}\|\mathbf{u}^{i-1} - \mathbf{x}_0^i\|_2^2\right)$
**end for**
**Output:** $\mathbf{x}_0$

---

### 3.4. Parameterization

In practice, we carefully design the parameterization of the required models, including the one-step generator for generative denoiser $G_\theta(\mathbf{y}_\sigma, \sigma, \mathbf{z})$, the score model $D_\phi(\mathbf{x}_t, t, \mathbf{y}_\sigma, \sigma)$ in Equation (13), and the discriminator $C_\psi(\mathbf{x}_t, t, \mathbf{y}_\sigma, \sigma)$ in Equation (15). The configurations are provided in Appendix B.1. The careful design not only effectively reuses knowledge from the teacher diffusion model but also leverages the inductive bias of preconditioning from (Karras et al., 2022), which significantly accelerates training while maintaining performance.

## 4. Plug-and-Play Probabilistic Inference with Generative Denoiser

This section focuses on sampling from an unnormalized target distribution defined by

$$\pi(\mathbf{x}_0) \propto q_{\text{data}}(\mathbf{x}_0)\exp\left(-\frac{1}{\beta}\mathcal{E}(\mathbf{x}_0)\right), \quad (17)$$

where $\mathcal{E}(\mathbf{x}_0)$ is a given energy function, and $\beta > 0$ controls the influence of the energy. Efficient sampling from $\pi(\mathbf{x}_0)$ is crucial for various machine learning tasks, including classical Bayesian inference where $\frac{1}{\beta}\mathcal{E}(\mathbf{x}_0)$ represents the negative log-likelihood, determining optimal policies in offline reinforcement learning (Peters et al., 2010; Lu et al., 2023), modeling desired sample distributions aligned with human preferences (Korbak et al., 2022; Uehara et al., 2025), among others. To address this, we propose a plug-and-play (PnP) probabilistic inference method, dubbed PnP-GD, using a well-trained generative denoiser $\mu_\theta(\mathbf{x}_0|\mathbf{y}_\sigma)$ that can achieve *asymptotic exact* sampling from $\pi(\mathbf{x}_0)$ under the ideal scenario $\mu_\theta(\mathbf{x}_0|\mathbf{y}_\sigma) = q(\mathbf{x}_0|\mathbf{y}_\sigma)$, as detailed below.

**Split Gibbs Sampler:** Motivated by asymptotic exact posterior sampling for diffusion models (Wu et al., 2024; Xu & Chi, 2024), we leverage *Split Gibbs Sampler (SGS)* (Vono

et al., 2019; Coeurdoux et al., 2024) to achieve plug-and-play probabilistic inference with $\mu_\theta(\mathbf{x}_0|\mathbf{y}_\sigma)$. Specifically, we first introduce an auxiliary random variable $\mathbf{u}$ and define the joint distribution of $\mathbf{x}_0$ and $\mathbf{u}$ as $\pi_\sigma(\mathbf{x}_0, \mathbf{u}) := \exp(\log q_{\text{data}}(\mathbf{x}_0) - \frac{1}{\beta}\mathcal{E}(\mathbf{u}) - \frac{1}{2\sigma^2}\|\mathbf{u} - \mathbf{x}_0\|_2^2)$, where $\sigma$ governs the strength of the penalty of the difference between $\mathbf{x}_0$ and $\mathbf{u}$. Notably, the marginal distribution of $\pi_\sigma(\mathbf{x}_0, \mathbf{u})$, $\pi_\sigma(\mathbf{x}_0)$, converges to our target distribution $\pi(\mathbf{x}_0)$ as $\sigma \to 0$ in terms of the total variation distance (Vono et al., 2019), resulting in asymptotically exact sampling from $\pi(\mathbf{x}_0)$.

To sample from $\pi_\sigma(\mathbf{x}_0, \mathbf{u})$, SGS alternately implements sampling from $\pi_\sigma(\mathbf{x}_0|\mathbf{u})$ and $\pi_\sigma(\mathbf{u}|\mathbf{x}_0)$ with an annealing decreasing value of $\sigma$, as follows:

$$\pi_\sigma(\mathbf{x}_0|\mathbf{u}) \propto \exp\left(\log q_{\text{data}}(\mathbf{x}_0) - \frac{1}{2\sigma^2}\|\mathbf{u} - \mathbf{x}_0\|_2^2\right), \quad (18)$$

$$\pi_\sigma(\mathbf{u}|\mathbf{x}_0) \propto \exp\left(-\frac{1}{\beta}\mathcal{E}(\mathbf{u}) - \frac{1}{2\sigma^2}\|\mathbf{u} - \mathbf{x}_0\|_2^2\right), \quad (19)$$

where Equations (18) and (19) are referred to as prior step and likelihood step, respectively. As the prior step is equivalent to sampling from $q(\mathbf{x}_0|\mathbf{y}_\sigma = \mathbf{u})$, it can be approximated by a well-trained generative denoiser with input $\mathbf{u}$ and noise level $\sigma$, i.e., $\mu_\theta(\mathbf{x}_0|\mathbf{y}_\sigma = \mathbf{u})$. The likelihood step is generally straightforward to sample, for instance, by implementing the *Unadjusted Langevin Algorithm (ULA)* (Welling & Teh, 2011), provided that $\mathcal{E}$ is differentiable.

Note that PnP-DM and PnP-DM are both built upon the foundation of SGS. The primary distinction lies in how the prior step is approximated. In PnP-DM, simulating the reverse diffusion process is required, which is not only computationally inefficient but also prone to irreducible discretization errors. In contrast, our approach significantly improves computational efficiency by requiring only one or a few NFEs for the prior step, while being free from any errors beyond those introduced by imperfect model training.

**ULA with Adaptive Step Size:** Theoretically, when the potential function in ULA is $L$-gradient Lipschitz, ULA provides performance guarantees if its step size is smaller than a constant proportioned to $L^{-1}$ (Durmus et al., 2019; Balasubramanian et al., 2022). For the likelihood step in Equation (19), it can be shown that the potential function, $\frac{1}{\beta}\mathcal{E}(\cdot) + \frac{1}{2\sigma^2}\|\cdot - \mathbf{x}_0\|^2$, is $(\beta^{-1}L + \sigma^{-2})$-gradient Lipschitz, provided that $\mathcal{E}(\cdot)$ is $L$-gradient Lipschitz. Based on this, we propose an adaptive step size $\gamma_\sigma$ for ULA that adapts to the current noise level $\sigma$ of PnP-GD as $\gamma_\sigma := C_1 \cdot (\beta^{-1}C_2 + \sigma^{-2})^{-1}$, where $C_1$ and $C_2$ are hyperparameters. As can be seen, the step size decreases monotonically as $\sigma$ is annealed towards zero, which aligns with intuition. Note that Zhang et al. (2024) also suggested reducing the step size as $\sigma$ decreases, and empirical results indicate the effectiveness of this approach. In the experiments, $C_1$ is fixed to 0.1, while $C_2$ is tuned for different probabilistic inference tasks.

**EMA Samples:** We observe that the vanilla implementation of PnP-GD produces generated samples with excessively sharp details. We hypothesize that this issue arises from the amplification of fine-grained details during the final steps of the PnP-GD process. To mitigate this problem, we propose averaging over multiple samples in the last MCMC chain of the PnP-GD process using an *exponential moving average (EMA)*. This approach approximately computes an average of a mode of the posterior distribution $\pi(\mathbf{x}_0)$, bringing the result closer to the posterior mean and thus reducing the sharpness of the generated samples.

The pseudocode for the PnP-GD procedure is provided in Algorithm 1. Additionally, the multi-step sampling approach discussed in Section 3.3 can also be applied to approximate the prior step.

## 5. Experiments

In this section, we evaluate proposed method by testing the distilled generative denoisers on **i)** class-conditional image generation on ImageNet-64×64 and ImageNet-512×512 datasets (Deng et al., 2009), and **ii)** plug-and-play inverse problem solving with PnP-GD on FFHQ-256×256 dataset (Karras et al., 2019). In Appendix B, we provide further details regarding the NCVSD training, class-conditional generation and inverse problem solving using PnP-GD. We include additional experimental results in Appendix C.

### 5.1. Image Generation

**Setup:** To test the generation performance of the proposed NCVSD, we follow the settings of EDM2 (Karras et al., 2024), ECM (Geng et al., 2025), and sCM (Lu & Song, 2025), to train and scale different sizes of models on ImageNet 64×64 and ImageNet 512×512 datasets (Deng et al., 2009). Specifically, we distill models of different sizes from EDM2 teachers, including NCVSD-S, NCVSD-M, NCVSD-L from EDM2-S, EDM2-M and EDM2-L, repectively. We standardize to Fréchet Inception Distance (FID) (Heusel et al., 2017) to measure the generation performance to compare different methods, following EDM2.

**Baselines:** We select consistency models (CM) (Song et al., 2023c) and its subsequent improvements (Geng et al., 2025; Lu & Song, 2025) as our primary comparisons. This is because models trained using NCVSD and CM exhibit very similar behaviors. Both approaches generate clean data from its noisy version of arbitrary noise levels, support multi-step generation to balance sample quality and sampling cost, and allow zero-shot controllable sampling. For instance, the original version of CM (Song et al., 2023c) already supports zero-shot image editing, and recently Tian et al. (2024) and Xu et al. (2024) have developed methods for zero-shot controllable sampling with consistency models to address a broader range of inverse problems.

*Table 1.* Sample quality on class-conditional ImageNet-64×64 and ImageNet-512×512. We report the number of function evaluations (NFE), Fréchet Inception Distance (FID), and the number of training iterations (#Iter).

**Class-Conditional ImageNet-64×64**

| Method | NFE↓ | FID↓ | #Iter↓ |
|---|---|---|---|
| *Teacher Diffusion Models* | | | |
| EDM2-S (Karras et al., 2024) | 63 | 1.58 | 1024k |
| EDM2-M (Karras et al., 2024) | 63 | 1.43 | 2048k |
| EDM2-L (Karras et al., 2024) | 63 | 1.33 | 1024k |
| EDM2-XL (Karras et al., 2024) | 63 | 1.33 | 640k |
| *Consistency Models* | | | |
| CD (Song et al., 2023c) | 2 | 4.70 | 600k |
| ECM-S (Geng et al., 2025) | 1 | 5.51 | 100k |
| | 2 | 3.18 | 100k |
| ECM-M (Geng et al., 2025) | 1 | 3.67 | 100k |
| | 2 | 2.35 | 100k |
| ECM-L (Geng et al., 2025) | 1 | 3.55 | 100k |
| | 2 | 2.14 | 100k |
| ECM-XL (Geng et al., 2025) | 1 | 3.35 | 100k |
| | 2 | 1.96 | 100k |
| sCD-XL (Lu & Song, 2025) | 1 | 2.44 | 400k |
| | 2 | 1.66 | 400k |
| *Noise Conditional Variational Score Distillation* | | | |
| NCVSD-S *(Proposed)* | 1 | 3.13 | 32k×3 |
| | 2 | 2.66 | 32k×3 |
| | 4 | 2.14 | 32k×3 |
| NCVSD-M *(Proposed)* | 1 | 3.04 | 32k×3 |
| | 2 | 2.47 | 32k×3 |
| | 4 | 1.92 | 32k×3 |
| NCVSD-L *(Proposed)* | 1 | 2.96 | 32k×3 |
| | 2 | 2.35 | 32k×3 |
| | 4 | 1.53 | 32k×3 |

**Class-Conditional ImageNet-512×512**

| Method | NFE↓ | FID↓ | #Iter↓ |
|---|---|---|---|
| *Teacher Diffusion Models* | | | |
| EDM2-S (Karras et al., 2024) | 63×2 | 2.23 | 2048k |
| EDM2-M (Karras et al., 2024) | 63×2 | 2.01 | 2048k |
| EDM2-L (Karras et al., 2024) | 63×2 | 1.88 | 1792k |
| EDM2-XL (Karras et al., 2024) | 63×2 | 1.85 | 1280k |
| EDM2-XXL (Karras et al., 2024) | 63×2 | 1.81 | 896k |
| *Consistency Models* | | | |
| sCD-S (Lu & Song, 2025) | 1 | 3.07 | 200k |
| | 2 | 2.50 | 200k |
| sCD-M (Lu & Song, 2025) | 1 | 2.75 | 200k |
| | 2 | 2.26 | 200k |
| sCD-L (Lu & Song, 2025) | 1 | 2.55 | 200k |
| | 2 | 2.04 | 200k |
| sCD-XL (Lu & Song, 2025) | 1 | 2.40 | 200k |
| | 2 | 1.93 | 200k |
| sCD-XXL (Lu & Song, 2025) | 1 | 2.28 | 200k |
| | 2 | 1.88 | 200k |
| *Noise Conditional Variational Score Distillation* | | | |
| NCVSD-S *(Proposed)* | 1 | 2.95 | 32k×3 |
| | 2 | 2.60 | 32k×3 |
| | 4 | 2.00 | 32k×3 |
| NCVSD-M *(Proposed)* | 1 | 2.85 | 32k×3 |
| | 2 | 2.08 | 32k×3 |
| | 4 | 1.92 | 32k×3 |
| NCVSD-L *(Proposed)* | 1 | 2.56 | 32k×3 |
| | 2 | 2.03 | 32k×3 |
| | 4 | 1.76 | 32k×3 |

**Results:** Table 1 compares various methods on class-conditional image generation using Fréchet Inception Distance (FID), Number of Function Evaluations (NFE), and training iterations. As can be seen, our methods outperform CMs of comparable sizes while requiring significantly fewer training iterations (32k×3 versus 200k, when accounting for score model and discriminator iterations for a fair comparison), demonstrating computational efficiency. Our methods also exhibit predictable performance gains with increased model size; the FID of NCVSD consistently decreases as model sizes grow, indicating training time scalability similar to that of teacher EDM2 models and CMs. Furthermore, by increasing test-time computation (i.e., NFE), our methods can match or exceed the performance of larger diffusion models or CMs. For instance, on the ImageNet-512×512 dataset, the FID of 4-step NCVSD-L (1.76) surpasses that of 2-step sCD-XXL (1.88) and the diffusion model EDM2-

XXL (1.81). Similar results are observed on the ImageNet-64×64 dataset, where the FID of 4-step NCVSD-L (1.53) is better than the FID of 2-step sCD-XL (1.66).

### 5.2. Inverse Problem Solving

**Setup:** To evaluate the zero-shot probabilistic inference ability, we test PnP-GD (Section 4) on several inverse problems on the FFHQ $256 \times 256$ dataset (Karras et al., 2019). To obtain a generative denoiser, we first pretrain a XS size EDM2 model on FFHQ $256 \times 256$ and then distill using NCVSD (see Appendix B.3 for details). The performance is evaluated using Learned Perceptual Image Patch Similarity (LPIPS) (Zhang et al., 2018) for perceptual quality, and Peak Signal-to-Noise Ratio (PSNR) for distortion quality.

We conduct comparisons across a variety of noisy linear and nonlinear inverse problems. For linear inverse problems, we

*Table 2.* Quantitative results on noisy inverse problems. The results are averaged over 100 images. We use **bold** and underline when the proposed method (PnP-GD) achieves the best and the second best, respectively.

| Method | NFE↓ | Inpaint (box) | | Deblur (Gaussian) | | Deblur (motion) | | Super resolution | | Phase retrieval | |
|---|---|---|---|---|---|---|---|---|---|---|---|
| | | LPIPS↓ | PSNR↑ | LPIPS↓ | PSNR↑ | LPIPS↓ | PSNR↑ | LPIPS↓ | PSNR↑ | LPIPS↓ | PSNR↑ |
| DDRM (Kawar et al., 2022) | 100 | 0.159 | 22.37 | 0.236 | 23.36 | - | - | 0.210 | 27.65 | - | - |
| DPS (Chung et al., 2023) | 1000 | 0.198 | 23.32 | 0.211 | 25.52 | 0.270 | 23.14 | 0.260 | 24.38 | 0.410 | 17.64 |
| DiffPIR (Zhu et al., 2023) | 100 | 0.186 | 25.02 | 0.236 | 27.36 | 0.255 | 26.57 | 0.260 | 26.64 | - | - |
| ΠGDM (Song et al., 2023a) | 99 | 0.284 | 21.76 | 0.245 | 25.72 | 0.240 | 26.29 | 0.245 | 25.69 | - | - |
| DWT-Var (Peng et al., 2024) | 99 | 0.158 | 21.26 | 0.186 | 27.70 | 0.189 | 28.06 | 0.187 | 27.78 | - | - |
| DAPS (Zhang et al., 2024) | 1000 | 0.133 | 24.07 | 0.165 | 29.19 | 0.157 | 29.66 | 0.177 | 29.07 | 0.140 | 29.94 |
| PnP-DM (Wu et al., 2024) | 2483 | - | - | 0.191 | 27.81 | 0.183 | 28.23 | 0.190 | 27.77 | 0.364 | 22.39 |
| PnP-GD (*Proposed*) | **50** | **0.128** | 21.75 | **0.155** | 27.06 | 0.160 | 28.02 | **0.151** | 27.93 | 0.186 | 27.82 |

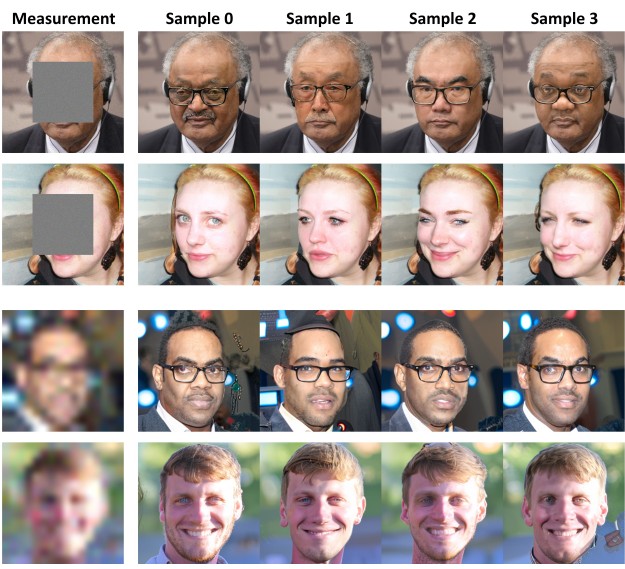

*Figure 2.* Sample diversity for addressing ill-poseness. Top: box inpainting with 128×128 mask. Bottom: super resolution from 16× downsampled images.

consider (1) box inpainting using a center box mask of size 128×128, (2) Gaussian deblurring with kernels sized 61×61 and a standard deviation of 3.0, and (3) motion deblurring with kernels sized 61 × 61 and a std of 0.5. For nonlinear inverse problems, we consider (1) super-resolution from 4×-bicubic downscaled images and (2) the challenging phase retrieval problem with 4× oversampling. Following (Chung et al., 2023; Zhang et al., 2024), we report the best result from four independent samples.

**Baselines:** We select the following baselines: (1) plug-and-play diffusion models (PnP-DM) (Wu et al., 2024), (2) decoupled annealing posterior sampling (DAPS) (Zhang et al., 2024), (3) guided diffusion models with learned wavelet variances (DWT-Var) (Peng et al., 2024), (4) pseudoinverse-guided diffusion models (ΠGDM) (Song et al., 2023a), (5) diffusion posterior sampling (DPS) (Chung et al., 2023), diffusion models for plug-and-play image restoration (Diff-PIR) (Zhu et al., 2023), and denoising diffusion restoration models (DDRM) (Kawar et al., 2022).

**Results:** The quantitative results for noisy inverse problems are presented in Table 2. Our method achieves the best or second best LPIPS across all inverse problems, demonstrating a superior perceptual quality compared to diffusion-based solvers. Notably, the state-of-the-art diffusion-based method, DAPS, requires 1k NFE, whereas our method uses only 50 NFE. We observed that the PSNR performance of our method does not achieve the best performance as LPIPS, which can be attributed to the *distortion-perception trade-off* (Blau & Michaeli, 2018), indicating that our method tends to generate results that retain more high-frequency details rather than approximate the mean of all possible solutions. This approach typically leads to higher MSE (lower PSNR) but aligns more closely with perceptual quality metrics such as LPIPS (Chung et al., 2023; Zhang et al., 2024). In Figure 2, we show that PnP-GD can generate images with diversity as well as fined details for addressing ill-poseness in inverse problems. Additionally, our approach efficiently and stably handles challenging nonlinear phase retrieval problem. While current diffusion-based solvers typically require over 1k NFE to achieve reasonable reconstructions for phase retrieval, our method attains competitive performance with just 50 NFE, representing roughly 20× acceleration.

## 6. Discussion

Our method integrates insights from multiple fields, refining and advancing existing approaches. Conceptually, the key distinction between our method and recent generative models—particularly diffusion models (DMs) and consistency models (CMs) (Song et al., 2023c) – lies in how clean data is predicted from its noisy counterpart. Our method

directly models the full posterior distribution over clean data, whereas DMs learn the MMSE prediction, and CMs solve for the initial conditions of PF-ODEs. This design choice offers notable advantages. Compared to DMs, our approach enables more efficient data generation. Meanwhile, it surpasses CMs in facilitating plug-and-play probabilistic inference via SGS, offering asymptotically exact guarantee.

Regarding multi-step sampling, our method requires significantly fewer NFEs to match the performance of DMs. This efficiency stems from modeling reverse transitions using *multi-modal implicit distributions* rather than single-modal Gaussian distributions. Our approach shares similarities with denoising diffusion GANs (Xiao et al., 2022) and moment matching (Salimans et al., 2024). However, denoising diffusion GANs are trained on a fixed, limited set of noise levels, restricting flexibility in inference-time control. Moment matching, on the other hand, does not function as a true denoising posterior sampler since its output remains deterministic with respect to the noisy input.

Another closely related line of work involves the distillation of diffusion models into amortized posterior samplers (Mammadov et al., 2024; Lee et al., 2025). Our approach offers significant advantages in inference-time flexibility, extending beyond the constraints of solving a single inverse problem defined at training. These advantages include generating high-quality *unconditional* samples and tackling a broader range of inverse problems. Furthermore, since prior terms are optimized using proxy objectives in these works (see Equation (14) in (Mammadov et al., 2024) and Equation (9) in (Lee et al., 2025)), they do not guarantee that the posterior distribution is the unique minimizer of their loss functions. Consequently, these methods cannot be directly applied to train generative denoisers that support marginal-preserving multi-step sampling (Section 3.3) and the SGS sampler (Section 4), as our method does.

## 7. Conclusion

In conclusion, we propose NCVSD, a novel method for distilling pretrained diffusion models into generative denoisers. NCVSD is grounded in the theoretical insight that the unconditional score function implicitly characterizes the score function of denoising posterior distributions. Empirically, our method exhibits outstanding performance in both few-step image generation and zero-shot inverse problem solving tasks, proving its potential as an efficient and flexible generative model.

**Limitations:** The proposed method relies on pre-trained diffusion models to distill a generative denoiser, which limits the possibility of training a denoiser from scratch. Additionally, achieving state-of-the-art performance requires adversarial training, which involves careful manual tuning

to ensure stable convergence. As a result, we have not been able to validate the effectiveness of NCVSD beyond its current scale due to computational constraints. We also note that the proposed noise conditional score estimator (Equation (12)) is not limited to the VSD framework; rather, it can be integrated with any score distillation method to distill generative denoisers from pretrained score models. Improving stability of distillation, developing pretraining techniques for generative denoisers, and applying the noise conditional score estimator to other score distillation methods all represent promising avenues for future research.

## Impact Statement

This work contributes to synthetic data generation and data augmentation by enabling the efficient production of high-quality samples. It also addresses a key challenge in diffusion-based inverse problem solvers by providing an efficient and accurate posterior sampling method, facilitating fast solutions across various scenarios. However, the potential misuse of synthetic data generation, such as creating harmful or misleading content, raises ethical concerns. Responsible deployment and adherence to ethical guidelines are essential to ensure the technology is used for societal benefit.

## Acknowledgements

This work was supported in part by the National Natural Science Foundation of China under Grant 62320106003, Grant U24A20251, Grant 62401357, Grant 62401366, Grant 62431017, Grant 62125109, Grant 62371288, Grant 62301299, Grant 62120106007, and in part by the Program of Shanghai Science and Technology Innovation Project under Grant 24BC3200800. The computations in this paper were partially run on the Baiyulan AI for Science Platform supported by the Artificial Intelligence Institute at Shanghai Jiao Tong University.

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

# A. Proofs

## A.1. Derivation of NCVSD Gradient as in Equation (9)

Denote $\mathbf{x}_t = G_\theta(\mathbf{y}_\sigma, \sigma, \mathbf{z}) + t \cdot \epsilon$, $\epsilon \sim \mathcal{N}(0, \mathbf{I})$, we have

$$
\begin{aligned}
\nabla_\theta &\mathbb{E}_{t,\mathbf{y}_\sigma} [D_{KL}(p_\theta(\mathbf{x}_t|\mathbf{y}_\sigma)||q(\mathbf{x}_t|\mathbf{y}_\sigma))] \\
&= \mathbb{E}_{t,\mathbf{y}_\sigma,\mathbf{z},\epsilon}[\nabla_\theta(\log p_\theta(\mathbf{x}_t|\mathbf{y}_\sigma) - \log q(\mathbf{x}_t|\mathbf{y}_\sigma))] \\
&= \mathbb{E}_{t,\mathbf{y}_\sigma,\mathbf{z},\epsilon}\left[\frac{\partial}{\partial\theta}\log p_\theta(\mathbf{x}_t|\mathbf{y}_\sigma) + \nabla_{\mathbf{x}_t}(\log p_\theta(\mathbf{x}_t|\mathbf{y}_\sigma) - \log q(\mathbf{x}_t|\mathbf{y}_\sigma))\frac{\partial\mathbf{x}_t}{\partial\theta}\right] \\
&= \underbrace{\mathbb{E}_{t,\mathbf{y}_\sigma,\mathbf{z},\epsilon}\left[\frac{\partial}{\partial\theta}\log p_\theta(\mathbf{x}_t|\mathbf{y}_\sigma)\right]}_{①} + \mathbb{E}_{t,\mathbf{y}_\sigma,\mathbf{z},\epsilon}\left[(\nabla_{\mathbf{x}_t}\log p_\theta(\mathbf{x}_t|\mathbf{y}_\sigma) - \nabla_{\mathbf{x}_t}\log q(\mathbf{x}_t|\mathbf{y}_\sigma))\frac{\partial G_\theta(\mathbf{y}_\sigma,\sigma,\mathbf{z})}{\partial\theta}\right].
\end{aligned}
\tag{20}
$$

It suffices to show that ① equals to zero:

$$
\begin{aligned}
\mathbb{E}_{t,\mathbf{y}_\sigma,\mathbf{z},\epsilon}\left[\frac{\partial}{\partial\theta}\log p_\theta(\mathbf{x}_t|\mathbf{y}_\sigma)\right] &= \mathbb{E}_{t,\mathbf{y}_\sigma}\left[\int p_\theta(\mathbf{x}_t|\mathbf{y}_\sigma)\frac{\partial}{\partial\theta}\log p_\theta(\mathbf{x}_t|\mathbf{y}_\sigma)\mathrm{d}\mathbf{x}_t\right] \\
&= \mathbb{E}_{t,\mathbf{y}_\sigma}\left[\int p_\theta(\mathbf{x}_t|\mathbf{y}_\sigma)\frac{1}{p_\theta(\mathbf{x}_t|\mathbf{y}_\sigma)}\frac{\partial}{\partial\theta}p_\theta(\mathbf{x}_t|\mathbf{y}_\sigma)\mathrm{d}\mathbf{x}_t\right] \\
&= \mathbb{E}_{t,\mathbf{y}_\sigma}\left[\frac{\partial}{\partial\theta}\int p_\theta(\mathbf{x}_t|\mathbf{y}_\sigma)\mathrm{d}\mathbf{x}_t\right] \\
&= \mathbb{E}_{t,\mathbf{y}_\sigma}\left[\frac{\partial}{\partial\theta}1\right] = 0.
\end{aligned}
\tag{21}
$$

## A.2. Noise Conditional Score Estimator

The Tweedie's formula in Equation (2) plays a central role in deriving the conditional score estimator for NCVSD. For completeness, we provide the proof of its conditional version here.

**Lemma 1** (Conditional Tweedie's formula). *If $\mathbf{x}_0, \mathbf{y}, \mathbf{x}_t$ follow the joint distribution $q(\mathbf{x}_0, \mathbf{y}, \mathbf{x}_t) = q(\mathbf{x}_0)q(\mathbf{y}|\mathbf{x}_0)q(\mathbf{x}_t|\mathbf{x}_0)$ with $q(\mathbf{x}_t|\mathbf{x}_0) = \mathcal{N}(\mathbf{x}_t|\mathbf{x}_0, t^2\mathbf{I})$. Then*

$$
\nabla_{\mathbf{x}_t}\log q(\mathbf{x}_t|\mathbf{y}) = t^{-2}\Big(\mathbb{E}[\mathbf{x}_0|\mathbf{x}_t,\mathbf{y}] - \mathbf{x}_t\Big).
$$

*Proof.*

$$
\begin{aligned}
\nabla_{\mathbf{x}_t}\log q(\mathbf{x}_t|\mathbf{y}) &= \frac{\nabla_{\mathbf{x}_t}q(\mathbf{x}_t|\mathbf{y})}{q(\mathbf{x}_t|\mathbf{y})} \\
&= \frac{1}{q(\mathbf{x}_t|\mathbf{y})}\nabla_{\mathbf{x}_t}\int q(\mathbf{x}_t|\mathbf{x}_0,\mathbf{y})q(\mathbf{x}_0|\mathbf{y})\mathrm{d}\mathbf{x}_0 \\
&= \frac{1}{q(\mathbf{x}_t|\mathbf{y})}\nabla_{\mathbf{x}_t}\int q(\mathbf{x}_t|\mathbf{x}_0)q(\mathbf{x}_0|\mathbf{y})\mathrm{d}\mathbf{x}_0 \quad \text{(by conditional independence } \mathbf{x}_t \perp\!\!\!\perp \mathbf{y} \mid \mathbf{x}_0) \\
&= \frac{1}{q(\mathbf{x}_t|\mathbf{y})}\int q(\mathbf{x}_0|\mathbf{y})\nabla_{\mathbf{x}_t}q(\mathbf{x}_t|\mathbf{x}_0)\mathrm{d}\mathbf{x}_0 \\
&= \frac{1}{q(\mathbf{x}_t|\mathbf{y})}\int q(\mathbf{x}_0|\mathbf{y})q(\mathbf{x}_t|\mathbf{x}_0)\nabla_{\mathbf{x}_t}\log q(\mathbf{x}_t|\mathbf{x}_0)\mathrm{d}\mathbf{x}_0 \\
&= \int q(\mathbf{x}_0|\mathbf{x}_t,\mathbf{y})\nabla_{\mathbf{x}_t}\log q(\mathbf{x}_t|\mathbf{x}_0)\mathrm{d}\mathbf{x}_0 \quad \text{(by Bayes' rule: } q(\mathbf{x}_0|\mathbf{x}_t,\mathbf{y}) = \tfrac{q(\mathbf{x}_0|\mathbf{y})q(\mathbf{x}_t|\mathbf{x}_0)}{q(\mathbf{x}_t|\mathbf{y})}) \\
&= \mathbb{E}_{q(\mathbf{x}_0|\mathbf{x}_t,\mathbf{y})}\left[t^{-2}(\mathbf{x}_0 - \mathbf{x}_t)\right] = t^{-2}\big(\mathbb{E}[\mathbf{x}_0|\mathbf{x}_t,\mathbf{y}] - \mathbf{x}_t\big).
\end{aligned}
\tag{22}
$$

$\square$

Then, we provide the proof of Proposition 1 in the main paper as follows.

**Proposition 1.** *Suppose* $(\mathbf{x}_0, \mathbf{y}_\sigma, \mathbf{x}_t)$ *follow the joint distribution* $q_{data}(\mathbf{x}_0)\mathcal{N}(\mathbf{y}_\sigma|\mathbf{x}_0, \sigma^2\mathbf{I})\mathcal{N}(\mathbf{x}_t|\mathbf{x}_0, t^2\mathbf{I})$. *For any* $\rho > 0$, *define the denoising posterior of* $\mathbf{x}_0$ *with noise level* $\rho$ *as* $q(\mathbf{x}_0|\mathbf{y}_\rho) \propto q_{data}(\mathbf{x}_0)\mathcal{N}(\mathbf{y}_\rho|\mathbf{x}_0, \rho^2\mathbf{I})$. *We obtain that*

$$q(\mathbf{x}_0|\mathbf{x}_t, \mathbf{y}_\sigma) = q\left(\mathbf{x}_0 \mid \mathbf{y}_{\sigma_{\text{eff}}} = \frac{\sigma^{-2}\mathbf{y}_\sigma + t^{-2}\mathbf{x}_t}{\sigma^{-2} + t^{-2}}\right),$$

$$\nabla_{\mathbf{x}_t} \log q(\mathbf{x}_t|\mathbf{y}_\sigma) = t^{-2}\left(\mathbb{E}\left[\mathbf{x}_0 \mid \mathbf{y}_{\sigma_{\text{eff}}} = \frac{\sigma^{-2}\mathbf{y}_\sigma + t^{-2}\mathbf{x}_t}{\sigma^{-2} + t^{-2}}\right] - \mathbf{x}_t\right),$$

*where* $\sigma_{\text{eff}} = (\sigma^{-2} + t^{-2})^{-\frac{1}{2}}$ *is the noise level of* $q(\mathbf{x}_0|\mathbf{y}_{\sigma_{\text{eff}}})$, *which is referred to as the effective noise level.*

*Proof.*

$$
\begin{aligned}
q(\mathbf{x}_0|\mathbf{x}_t, \mathbf{y}_\sigma) &\propto q_{\text{data}}(\mathbf{x}_0)q(\mathbf{x}_t|\mathbf{x}_0)q(\mathbf{y}_\sigma|\mathbf{x}_0) \\
&\propto q_{\text{data}}(\mathbf{x}_0)\exp\left(-\frac{1}{2t^2}\|\mathbf{x}_0 - \mathbf{x}_t\|^2\right)\exp\left(-\frac{1}{2\sigma^2}\|\mathbf{x}_0 - \mathbf{y}_\sigma\|^2\right) \\
&\propto q_{\text{data}}(\mathbf{x}_0)\exp\left(-\left(\frac{1}{2t^2} + \frac{1}{2\sigma^2}\right)\|\mathbf{x}_0\|^2 + \langle\mathbf{x}_0, t^{-2}\mathbf{x}_t + \sigma^{-2}\mathbf{y}_\sigma\rangle\right) \\
&\propto q_{\text{data}}(\mathbf{x}_0)\exp\left(-\frac{1}{2(t^{-2}+\sigma^{-2})^{-1}}\left(\|\mathbf{x}_0\|^2 - 2\langle\mathbf{x}_0, \frac{t^{-2}\mathbf{x}_t + \sigma^{-2}\mathbf{y}_\sigma}{t^{-2} + \sigma^{-2}}\rangle\right)\right) \\
&\propto q_{\text{data}}(\mathbf{x}_0)\exp\left(-\frac{1}{2(t^{-2}+\sigma^{-2})^{-1}}\left\|\mathbf{x}_0 - \frac{t^{-2}\mathbf{x}_t + \sigma^{-2}\mathbf{y}_\sigma}{t^{-2} + \sigma^{-2}}\right\|^2\right) \quad \text{(completing the square)} \\
&\propto q\left(\mathbf{x}_0 \mid \mathbf{y}_{\sigma_{\text{eff}}} = \frac{\sigma^{-2}\mathbf{y}_\sigma + t^{-2}\mathbf{x}_t}{\sigma^{-2} + t^{-2}}\right).
\end{aligned}
\tag{23}
$$

Both sides are normalized, so Equation (10) holds.

By Lemma 1, we have $\nabla_{\mathbf{x}_t}\log q(\mathbf{x}_t|\mathbf{y}_\sigma) = t^{-2}(\mathbb{E}[\mathbf{x}_0|\mathbf{x}_t, \mathbf{y}_\sigma] - \mathbf{x}_t)$. Additionally, Equation (23) implies $\mathbb{E}[\mathbf{x}_0|\mathbf{x}_t, \mathbf{y}_\sigma] = \mathbb{E}[\mathbf{x}_0 \mid \mathbf{y}_{\sigma_{\text{eff}}} = \frac{\sigma^{-2}\mathbf{y}_\sigma + t^{-2}\mathbf{x}_t}{\sigma^{-2} + t^{-2}}]$. Hence, Equation (11) holds. $\square$

## A.3. Multi-Step Sampling

NCVSD implements multi-step sampling based on a DDIM-like latent variable model, and we provide the pseudo code in Algorithm 2. In this section, we provide a detailed definition of the latent variable model and demonstrate that it correctly preserves desired marginals. For ease of reading, we rewrite the definition as follows:

$$q(\mathbf{x}_{0:N}|\mathbf{y}_\sigma) = q(\mathbf{x}_0|\mathbf{y}_\sigma)q(\mathbf{x}_{1:N}|\mathbf{x}_0), \tag{24}$$

$$q(\mathbf{x}_{1:N}|\mathbf{x}_0) = q(\mathbf{x}_N|\mathbf{x}_0)\prod_{i=2}^{N} q(\mathbf{x}_{i-1}|\mathbf{x}_i, \mathbf{x}_0), \tag{25}$$

where $q(\mathbf{x}_N|\mathbf{x}_0)$ is defined as $\mathcal{N}(\mathbf{x}_0, \sigma_N^2\mathbf{I})$. To ensure that the marginal distribution $q(\mathbf{x}_i|\mathbf{x}_0)$ equals to $\mathcal{N}(\mathbf{x}_0, \sigma_i^2\mathbf{I})$ for all $i = 1, 2, ..., N$, we construct $q(\mathbf{x}_{i-1}|\mathbf{x}_i, \mathbf{x}_0)$ as follows:

$$\mathbf{x}_{i-1} = \mathbf{x}_0 + \sigma_{i-1}\left(\sqrt{\zeta}\epsilon + \sqrt{1-\zeta}\hat{\epsilon}\right), \ \epsilon \sim \mathcal{N}(0, \mathbf{I}), \ \hat{\epsilon} = \frac{\mathbf{x}_i - \mathbf{x}_0}{\sigma_i}, \tag{26}$$

where $\zeta \in [0, 1]$. Or equivalently,

$$q(\mathbf{x}_{i-1}|\mathbf{x}_i, \mathbf{x}_0) := \mathcal{N}\left(\mathbf{x}_0 + \sigma_{i-1}\sqrt{1-\zeta} \cdot \frac{\mathbf{x}_i - \mathbf{x}_0}{\sigma_i}, \sigma_{i-1}^2\zeta\mathbf{I}\right). \tag{27}$$

Then, we provide the proof of Proposition 2 in the main paper as follows.

**Proposition 2.** *By constructing the following distributions:*

$$
\begin{aligned}
q(\mathbf{x}_N|\mathbf{x}_0) &= \mathcal{N}(\mathbf{x}_0, \sigma_N^2\mathbf{I}), \\
q(\mathbf{x}_{i-1}|\mathbf{x}_i, \mathbf{x}_0) &= \mathcal{N}\left(\mathbf{x}_0 + \sigma_{i-1}\sqrt{1-\zeta} \cdot \frac{\mathbf{x}_i - \mathbf{x}_0}{\sigma_i}, \sigma_{i-1}^2\zeta\mathbf{I}\right), \\
p(\mathbf{x}_N) &= \mathbb{E}_{q(\mathbf{x}_0|\mathbf{y}_\sigma)}[q(\mathbf{x}_N|\mathbf{x}_0)], \\
p(\mathbf{x}_{i-1}|\mathbf{x}_i) &= \mathbb{E}_{q(\mathbf{x}_0|\mathbf{x}_i, \mathbf{y}_\sigma)}[q(\mathbf{x}_{i-1}|\mathbf{x}_i, \mathbf{x}_0)],
\end{aligned}
$$

**Algorithm 2** Multi-step generative denoising

---

**Input:** noisy data $\mathbf{y}_\sigma$, input noise level $\sigma$, generative denoiser $\mu_\theta(\mathbf{x}_0|\mathbf{y}_\sigma)$, noise annealing schedule $\sigma_N > \sigma_{N-1} > ... > \sigma_1 \approx 0$, random factor $\zeta \in [0, 1]$

$\mathbf{x}_N \sim \mathcal{N}(\mathbf{0}, \sigma_N^2\mathbf{I})$

**for** $i = N, ..., 2$ **do**

$\quad \sigma_{\text{eff}} = (\sigma^{-2} + \sigma_i^{-2})^{-\frac{1}{2}}$

$\quad \mathbf{y}_{\sigma_{\text{eff}}} = \sigma_{\text{eff}}^2 \cdot (\sigma^{-2}\mathbf{y}_\sigma + \sigma_i^{-2}\mathbf{x}_i)$

$\quad \mathbf{x}_0 \sim \mu_\theta(\mathbf{x}_0|\mathbf{y}_{\sigma_{\text{eff}}})$

$\quad \mathbf{x}_{i-1} \sim q(\mathbf{x}_{i-1}|\mathbf{x}_i, \mathbf{x}_0)$

**end for**

**Output:** $\mathbf{x}_0$

---

*where $\zeta \in [0, 1]$. Then, we have that $q(\mathbf{x}_i|\mathbf{x}_0) = \mathcal{N}(\mathbf{x}_i|\mathbf{x}_0, \sigma_i^2\mathbf{I})$ and $p(\mathbf{x}_i) = q(\mathbf{x}_i|\mathbf{y}_\sigma)$ for $i = 1, 2, ..., N$. In addition, the following equality holds:*

$$q(\mathbf{x}_0|\mathbf{x}_i, \mathbf{y}_\sigma) = q\left(\mathbf{x}_0 \mid \mathbf{y}_{\sigma_{\text{eff}}} = \tfrac{\sigma^{-2}\mathbf{y}_\sigma + \sigma_i^{-2}\mathbf{x}_i}{\sigma^{-2} + \sigma_i^{-2}}\right),$$

*and the effective noise level $\sigma_{\text{eff}} = (\sigma^{-2} + \sigma_i^{-2})^{-\frac{1}{2}}$.*

*Proof.* We divide the proof into three parts, respectively devoted for proving $q(\mathbf{x}_i|\mathbf{x}_0) = \mathcal{N}(\mathbf{x}_0, \sigma_i^2\mathbf{I})$, $p(\mathbf{x}_i) = q(\mathbf{x}_i|\mathbf{y}_\sigma)$, and $q(\mathbf{x}_0|\mathbf{x}_i, \mathbf{y}_\sigma) = q\left(\mathbf{x}_0 \mid \mathbf{y}_{\sigma_{\text{eff}}} = \tfrac{\sigma^{-2}\mathbf{y}_\sigma + \sigma_i^{-2}\mathbf{x}_i}{\sigma^{-2} + \sigma_i^{-2}}\right)$.

**Part I:** Similar to (Lemma 1, Song et al. (2021a)), since $q(\mathbf{x}_i|\mathbf{x}_0) = \mathcal{N}(\mathbf{x}_0, \sigma_i^2\mathbf{I})$ already hold for $i = N$, we can prove that $q(\mathbf{x}_i|\mathbf{x}_0) = \mathcal{N}(\mathbf{x}_0, \sigma_i^2\mathbf{I})$ holds for $i = 1, 2, ..., N-1$ by induction. Specifically, suppose $q(\mathbf{x}_i|\mathbf{x}_0) = \mathcal{N}(\mathbf{x}_0, \sigma_i^2\mathbf{I})$, then

$$q(\mathbf{x}_{i-1}|\mathbf{x}_0) = \int q(\mathbf{x}_{i-1}|\mathbf{x}_i, \mathbf{x}_0)q(\mathbf{x}_i|\mathbf{x}_0)\mathrm{d}\mathbf{x}_i \tag{28}$$

is a Gaussian distribution. Its mean and variance can be determined by Bayes' theorem for Gaussian variables (2.115, Bishop & Nasrabadi (2006)) as

$$q(\mathbf{x}_{i-1}|\mathbf{x}_0) = \mathcal{N}\left(\mathbf{x}_0 + \sigma_{i-1}\sqrt{1-\zeta} \cdot \tfrac{\mathbf{x}_0 - \mathbf{x}_0}{\sigma_i}, \sigma_{i-1}^2\zeta\mathbf{I} + (\tfrac{\sigma_{i-1}\sqrt{1-\zeta}}{\sigma_i})^2 \cdot \sigma_i^2\mathbf{I}\right) = \mathcal{N}(\mathbf{x}_0, \sigma_{i-1}^2\mathbf{I}). \tag{29}$$

Therefore, $q(\mathbf{x}_i|\mathbf{x}_0) = \mathcal{N}(\mathbf{x}_0, \sigma_i^2\mathbf{I})$ for $i = 1, 2, ..., N$.

**Part II:** Since $p(\mathbf{x}_N) = \mathbb{E}_{q(\mathbf{x}_0|\mathbf{y}_\sigma)}[q(\mathbf{x}_N|\mathbf{x}_0)] = q(\mathbf{x}_N|\mathbf{y}_\sigma)$ already hold for $i = N$, we also prove the statement $p(\mathbf{x}_i) = q(\mathbf{x}_i|\mathbf{y}_\sigma)$ holds for $i = 0, 1, ..., N$ by induction. Specifically, suppose $p(\mathbf{x}_i) = q(\mathbf{x}_i|\mathbf{y}_\sigma)$, and we consider the following marginalization representation of $q(\mathbf{x}_{i-1}|\mathbf{y}_\sigma)$:

$$\begin{aligned}
q(\mathbf{x}_{i-1}|\mathbf{y}_\sigma) &= \int\int q(\mathbf{x}_{i-1}, \mathbf{x}_0, \mathbf{x}_i|\mathbf{y}_\sigma)\mathrm{d}\mathbf{x}_0\mathrm{d}\mathbf{x}_i \\
&= \int\int q(\mathbf{x}_i|\mathbf{y}_\sigma)q(\mathbf{x}_0|\mathbf{x}_i, \mathbf{y}_\sigma)q(\mathbf{x}_{i-1}|\mathbf{x}_0, \mathbf{x}_i, \mathbf{y}_\sigma)\mathrm{d}\mathbf{x}_0\mathrm{d}\mathbf{x}_i \\
&= \int \underbrace{q(\mathbf{x}_i|\mathbf{y}_\sigma)}_{p(\mathbf{x}_i)}\underbrace{\left(\int q(\mathbf{x}_0|\mathbf{x}_i, \mathbf{y}_\sigma)q(\mathbf{x}_{i-1}|\mathbf{x}_0, \mathbf{x}_i)\mathrm{d}\mathbf{x}_0\right)}_{p(\mathbf{x}_{i-1}|\mathbf{x}_i)}\mathrm{d}\mathbf{x}_i \\
&= p(\mathbf{x}_{i-1}). \tag{30}
\end{aligned}$$

Therefore, $p(\mathbf{x}_i) = q(\mathbf{x}_i|\mathbf{y}_\sigma)$ for $i = 0, 1, ..., N$.

**Part III:** By $q(\mathbf{x}_i|\mathbf{x}_0) = \mathcal{N}(\mathbf{x}_0, \sigma_i^2\mathbf{I})$, the joint distribution of $\mathbf{x}_0, \mathbf{y}_\sigma, \mathbf{x}_i$ is $q_{\text{data}}(\mathbf{x}_0)\mathcal{N}(\mathbf{y}_\sigma|\mathbf{x}_0, \sigma^2\mathbf{I})\mathcal{N}(\mathbf{x}_i|\mathbf{x}_0, \sigma_i^2\mathbf{I})$. Equation (16) holds according to Proposition 1. Thus, we conclude the proof. $\square$

---

**Algorithm 3** One gradient optimization step of the generative denoiser $G_\theta$

---

**Input:** generative denoiser $G_\theta$ and EMA parameter $\theta^-$, score model $D_0$ for estimating data score $\nabla \log q(\mathbf{x}_t|\mathbf{y}_\sigma)$, score model $D_\phi$ for estimating model score $\nabla \log p_\theta(\mathbf{x}_t|\mathbf{y}_\sigma)$, uncertainty weighting model $w_\lambda$, discriminator $C_\psi$, learning rate $\eta$, and EMA rate function $\beta$

Sample $\mathbf{x}_0$ from the dataset
Sample $t, \sigma$ from predefined distributions
Sample $\mathbf{y}_\sigma \sim \mathcal{N}(\mathbf{x}_0, \sigma^2 \mathbf{I})$
Sample $\mathbf{x}_\theta = G_\theta(\mathbf{y}_\sigma, \sigma, \mathbf{z})$, where $\mathbf{z} \sim \mathcal{N}(\mathbf{0}, \mathbf{I})$
Sample $\tilde{\mathbf{x}}_\theta \sim \mathcal{N}(\mathbf{x}_\theta, t^2 \mathbf{I})$

Compute effective noisy sample as

$$\sigma_{\text{eff}} \leftarrow (\sigma^{-2} + t^{-2})^{-\frac{1}{2}}, \ \mathbf{y}_{\sigma_{\text{eff}}} \leftarrow \sigma_{\text{eff}}^2 \cdot (\sigma^{-2} \mathbf{y}_\sigma + t^{-2} \tilde{\mathbf{x}}_\theta)$$

Compute estimation for the data and model scores (rescale and omit $\mathbf{x}_t$)

$$\mathbf{s}_0 \leftarrow D_0(\mathbf{y}_{\sigma_{\text{eff}}}, \sigma_{\text{eff}}), \ \mathbf{s}_\phi \leftarrow D_\phi(\tilde{\mathbf{x}}_\theta, t, \mathbf{y}_\sigma, \sigma)$$

Compute loss and do gradient update for $\theta$

$$\mathcal{L}(\theta, \lambda) \leftarrow e^{-w_\lambda(t)} \|\mathbf{x}_\theta - \text{stopgrad}(\mathbf{s}_0 - \mathbf{s}_\phi + \mathbf{x}_\theta)\|_2^2 + \dim(\mathbf{x}_0) \cdot w_\lambda(t) - \dim(\mathbf{x}_0) \cdot \log C_\psi(\tilde{\mathbf{x}}_\theta, t, \mathbf{y}_\sigma, \sigma)$$
$$[\theta, \lambda] \leftarrow [\theta, \lambda] - \eta \cdot [\nabla_\theta \mathcal{L}(\theta, \lambda), \nabla_\lambda \mathcal{L}(\theta, \lambda)]$$
$$\theta^- \leftarrow \beta(t)\theta^- + (1 - \beta(t))\theta$$

---

## B. Experimental Details

In this section, we introduce parameterization of the neural networks used in the experiments, the training and inference details for NCVSD, and the details for inverse problem solving using PnP-GD.

### B.1. Parameterization

**Score model**  We utilize a score model $D_\phi$ to estimate the conditional score function $\nabla \log p_\theta(\mathbf{x}_t|\mathbf{y}_\sigma)$ in Equation (9):

$$\nabla \log p_\theta(\mathbf{x}_t|\mathbf{y}_\sigma) \approx t^{-2} \left(D_\phi(\mathbf{x}_t, t, \{\mathbf{y}_\sigma, \sigma\}) - \mathbf{x}_t\right), \tag{31}$$

where the parameters $\phi$ of $D_\phi$ are initialized from the teacher model $D_0$ and then fine-tuned via optimization of Equation (13) during the training process of the generator $G_\theta$. The additional condition inputs $(\mathbf{y}, \sigma)$ is injected into $D_\phi$ using a trainable *control net* (Zhang et al., 2023) like architecture, and we use $\{\cdot\}$ to emphasize its input. Specifically, we copy the encoder of the UNet model (does not share weights) and add the outputs of the copied encoder on the outputs of the original encoders before input into the UNet decoder. We discard the zero convolutions in the original control net since it will break the magnitude preserving property introduced by EDM2 (Karras et al., 2024). Instead, we propose a *learnable* magnitude preserving addition layer to achieve the same goal as the zero-convolutions. Specifically, denote $\mathbf{a}$ as the output of the original encoder, and denote $\mathbf{b}$ as the output of the copied encoder, we obtain the new output according to:

$$\text{MP-Sum}_w(\mathbf{a}, \mathbf{b}) := \frac{(1 - w)\mathbf{a} + w\mathbf{b}}{\sqrt{(1 - w)^2 + w^2}}, \tag{32}$$

where $w \in [0, 1]$ is a learnable weight, and is initialized to 0 to prevent disrupting knowledge of the pretrained model, which function similarly to the zero convolutions in the original control net (Zhang et al., 2023). We force $w$ lies in $[0, 1]$ using similar implementation to the forced weight normalization in EDM2.

**Generative denoiser**  In practice, we use a model $G_\theta(\mathbf{y}_\sigma, \sigma, \mathbf{z})$ to implement the generative denoiser $\mu_\theta(\mathbf{x}_0|\mathbf{y}_\sigma)$. We parameterize $G_\theta(\mathbf{y}_\sigma, \sigma, \mathbf{z})$ by adapting the network architecture of the pretrained score model $D_0(\mathbf{x}_t, t)$, given by

$$G_\theta(\mathbf{y}_\sigma, \sigma, \mathbf{z}) := D_\theta(\mathbf{y}_\sigma, \sigma, \{\mathbf{z}\}), \ \mathbf{z} \sim \mathcal{N}(\mathbf{0}, \mathbf{I}), \tag{33}$$

where $\mathbf{z}$ is an additional noise input to introduce stochasticity for generating random samples. The parameters $\theta$ of $D_\theta$ are initialized from the teacher model $D_0$, which enables efficient transfer of knowledge from the teacher model $D_0$, as well as reusing the inductive bias of preconditioning (Karras et al., 2022). However, we observed that directly using Equation (33) leads to severe mode collapse. To address this issue, we propose introducing stochasticity directly into $\mathbf{y}_\sigma$ by adding random noise $\mathbf{z}$. Specifically, we add $\mathbf{z}$ to $\mathbf{y}_\sigma$ using a scaling factor $\gamma$ to achieve a higher noise level $\hat{\sigma} = \sigma + \gamma\sigma$, resulting in $\mathbf{y}_{\hat{\sigma}} = \mathbf{y}_\sigma + \sqrt{\hat{\sigma}^2 - \sigma^2}\mathbf{z}$ [3]. Additionally, the original conditions $\mathbf{y}_\sigma$ and $\sigma$ are fed into a trainable ControlNet, similar to the score model $D_\phi$, to preserve critical information. Therefore, the generative denoiser is finally defined as:

$$G_\theta(\mathbf{y}_\sigma, \sigma, \mathbf{z}) := D_\theta(\mathbf{y}_{\hat{\sigma}}, \hat{\sigma}, \{\mathbf{y}_\sigma, \sigma\}). \tag{34}$$

**Discriminator**   To parameterize $C_\psi(\mathbf{x}_t, t, \mathbf{y}_\sigma, \sigma)$ for the adversarial loss in Equation (15), we employ two UNet encoders, $\mathcal{E}_{\psi_1}$ and $\mathcal{E}_{\psi_2}$, to extract features from $(\mathbf{x}_t, t)$ and $(\mathbf{y}_\sigma, \sigma)$, respectively. The outputs of the final layers of these encoders are concatenated along the channel dimension, followed by an average pooling layer applied to the spatial dimensions, a linear layer, and a sigmoid layer to produce a scalar output in [0, 1]:

$$C_\psi(\mathbf{x}_t, t, \mathbf{y}_\sigma, \sigma) := \mathrm{Proj}\Big(\mathcal{E}_{\psi_1}(\mathbf{x}_t, t), \mathcal{E}_{\psi_2}(\mathbf{y}_\sigma, \sigma)\Big), \tag{35}$$

where $\mathrm{Proj} := \mathrm{Sigmoid} \circ \mathrm{Linear} \circ \mathrm{AvgPool} \circ \mathrm{Concat}$, $\psi := [\psi_1, \psi_2]$, and $\mathcal{E}_{\psi_1}, \mathcal{E}_{\psi_2}$ are initialized from the encoder of the teacher model $D_0$.

### B.2. Training and Inference

**Uncertainty weighting**   We employ *uncertainty weighting* (Kendall et al., 2018; Karras et al., 2024; Lu & Song, 2025) to balance the loss contributions across different $t$. Specifically, since the NCVSD gradient is a vector-Jacobian product, we can convert it to a gradient of a L2 loss, as follows:

$$\nabla_\theta \mathcal{L}_{\mathrm{ncvsd}}(\theta) = \mathbb{E}\left[(\nabla_{\mathbf{x}_t} \log p_\theta(\mathbf{x}_t|\mathbf{y}_\sigma) - \nabla_{\mathbf{x}_t} \log q(\mathbf{x}_t|\mathbf{y}_\sigma))\frac{\partial G_\theta(\mathbf{y}_\sigma, \sigma, \mathbf{z})}{\partial \theta}\right] \tag{36}$$

$$\approx t^{-2}\mathbb{E}\left[(D_\phi(\mathbf{x}_t, t, \mathbf{y}_\sigma, \sigma) - D_0(\mathbf{y}_{\sigma_{\mathrm{eff}}}, \sigma_{\mathrm{eff}}))\frac{\partial G_\theta(\mathbf{y}_\sigma, \sigma, \mathbf{z})}{\partial \theta}\right] \tag{37}$$

$$= t^{-2}\nabla_\theta \mathbb{E}\Big[\|G_\theta(\mathbf{y}_\sigma, \sigma, \mathbf{z}) - \underbrace{\mathrm{stopgrad}(G_\theta(\mathbf{y}_\sigma, \sigma, \mathbf{z}) - D_\phi(\mathbf{x}_t, t, \mathbf{y}_\sigma, \sigma) + D_0(\mathbf{y}_{\sigma_{\mathrm{eff}}}, \sigma_{\mathrm{eff}}))}_{\text{L2 target}}\|_2^2\Big] \tag{38}$$

where $\mathrm{stopgrad}$ is the stop gradient operator, and the expectation is taken over $(t, \sigma, \mathbf{y}_\sigma, \mathbf{z}, \mathbf{x}_t)$ where $\mathbf{x}_t \sim \mathcal{N}(G_\theta(\mathbf{y}_\sigma, \sigma, \mathbf{z}), t^2\mathbf{I})$. In Equation (37), we leverage the teacher model $D_0$ and the score model $D_\phi$ to approximate the score functions as $\nabla_{\mathbf{x}_t} \log p_\theta(\mathbf{x}_t|\mathbf{y}_\sigma) \approx t^{-2}(D_\phi(\mathbf{x}_t, t, \mathbf{y}_\sigma, \sigma) - \mathbf{x}_t)$ and $\nabla_{\mathbf{x}_t} \log q(\mathbf{x}_t|\mathbf{y}_\sigma) \approx t^{-2}(D_0(\mathbf{y}_{\sigma_{\mathrm{eff}}}, \sigma_{\mathrm{eff}}) - \mathbf{x}_t)$. The scale of Equation (38) varies considerably across noise levels. To address this, we introduce an uncertainty weighting network $w_\lambda$ to ensure that the losses have unit variance accross noise levels. Specifically, we define the following two losses $\mathcal{L}_1$ and $\mathcal{L}_2$, which respectively provide unbiased estimates of the gradients of Equation (38) and $\mathcal{L}_{\mathrm{adv}}$ up to a scaling factor:

$$\mathcal{L}_1 := e^{-w_\lambda(t)}\|\mathbf{x}_\theta - \mathrm{stopgrad}(\mathbf{s}_0 - \mathbf{s}_\phi + \mathbf{x}_\theta)\|_2^2 + \dim(\mathbf{x}_\theta) \cdot w_\lambda(t) \tag{39}$$

$$\mathcal{L}_2 := -\log C_\psi(\tilde{\mathbf{x}}_\theta, t, \mathbf{y}_\sigma, \sigma) \tag{40}$$

where $\mathbf{x}_\theta := G_\theta(\mathbf{y}_\sigma, \sigma, \mathbf{z})$, $\tilde{\mathbf{x}}_\theta := \mathbf{x}_t$, $\mathbf{s}_0 := D_0(\mathbf{y}_{\sigma_{\mathrm{eff}}}, \sigma_{\mathrm{eff}})$, and $\mathbf{s}_\phi := D_\phi(\mathbf{x}_t, t, \mathbf{y}_\sigma, \sigma)$ for notation simplicity and to emphasize the dependency on parameters. Note that $\mathcal{L}_1$ can be viewed as the sum of independent negative Gaussian log-likelihoods over the data dimensions, while $\mathcal{L}_2$ is also a negative log-likelihood. To balance the contributions of the two losses, it is natural to scale $\mathcal{L}_2$ by the number of data dimensions. Therefore, we define the final loss as

$$\mathcal{L} := \mathcal{L}_1 + \dim(\mathbf{x}_\theta) \cdot \mathcal{L}_2. \tag{41}$$

To summarize, we provide pseudo code for the NCVSD training algorithm in Algorithm 3. Note that before optimizing $G_\theta$ using Algorithm 3, the score model $D_\phi$ and the discriminator $C_\psi$ should also be optimized for the distribution of $\mathbf{x}_0 = G_\theta(\mathbf{y}_\sigma, \sigma, \mathbf{z})$, as will be elaborated below.

---

[3]A similar approach has been proposed in EDM (Karras et al., 2022) for introducing stochasticity in the diffusion sampler.

*Table 3.* Hyperparameter details for training and inference.

| | ImageNet 64×64 | | | ImageNet 512×512 | | | FFHQ 256×256 |
| --- | --- | --- | --- | --- | --- | --- | --- |
| | S | M | L | S | M | L | XS |
| **Model details** | | | | | | | |
| Channel multiplier | 192 | 256 | 320 | 192 | 256 | 320 | 128 |
| Dropout probability | 0% | 0% | 0% | 0% | 0% | 0% | 0% |
| Stochasticity strength $\gamma$ | 0.414 | 0.414 | 0.414 | 0.414 | 0.414 | 0.414 | 0.414 |
| Model capacity of $G_\theta$ (Mparams) | 368.2 | 653.5 | 1020.1 | 368.2 | 653.5 | 1020.1 | 146.2 |
| **Training details** | | | | | | | |
| Effective batch size | 2048 | 2048 | 2048 | 2048 | 2048 | 2048 | 128 |
| Learning rate max ($\alpha_{\mathrm{ref}}$) | 0.0100 | 0.0090 | 0.0080 | 0.0100 | 0.0090 | 0.0080 | 0.0120 |
| Learning rate decay ($t_{\mathrm{ref}}$) | 35000 | 35000 | 70000 | 70000 | 70000 | 70000 | 35000 |
| Learning rate warm up $K$ images | 1000 | 1000 | 1000 | 1000 | 1000 | 1000 | 100 |
| Adversarial loss warm up $K$ images | 16778 | 16778 | 16778 | 16778 | 16778 | 16778 | 2097 |
| Learning rate scaling for $C_\psi$ | 0.01 | 0.01 | 0.01 | 0.01 | 0.01 | 0.01 | 0.01 |
| Learning rate scaling for $G_\theta$ | 0.01 | 0.01 | 0.01 | 0.01 | 0.01 | 0.01 | 0.01 |
| Adam $\beta_1$ | 0.9 | 0.9 | 0.9 | 0.9 | 0.9 | 0.9 | 0.9 |
| Adam $\beta_2$ | 0.99 | 0.99 | 0.99 | 0.99 | 0.99 | 0.99 | 0.99 |
| Training samples (Mi, $2^{20}$) | 64 | 64 | 64 | 64 | 64 | 64 | 4 |
| Noise distribution $P_{\mathrm{mean}}$ for $t$ | -0.8 | -0.8 | -0.8 | -0.4 | -0.4 | -0.4 | -0.8 |
| Noise distribution $P_{\mathrm{std}}$ for $t$ | 1.6 | 1.6 | 1.6 | 1.0 | 1.0 | 1.0 | 1.6 |
| **Training cost** | | | | | | | |
| Mixed precision | fp16 | fp16 | fp16 | fp16 | fp16 | fp16 | fp16 |
| Loss scaling to prevent overflows | 1.0 | 1.0 | 1.0 | 1.0 | 1.0 | 1.0 | 0.1 |
| Batch size per GPU | 64 | 32 | 16 | 64 | 32 | 8 | 8 |
| A100 GPU hours | ∼650 | ∼1100 | ∼1450 | ∼650 | ∼1100 | ∼1450 | ∼300 |
| **Inference details** | | | | | | | |
| Random factor $\zeta$ | 1.0 | 1.0 | 1.0 | 1.0 | 1.0 | 1.0 | - |
| 1-step sampling timesteps | 10 | 10 | 10 | 12 | 12 | 12 | - |
| 2-step sampling timesteps | 10,22 | 10,22 | 10,22 | 10,22 | 10,22 | 10,22 | - |
| 4-step sampling timesteps | 0,10,20,30 | 0,10,20,30 | 0,10,20,30 | 0,10,20,30 | 0,10,20,30 | 0,10,20,30 | - |

**Training hyperparameters** We perform one gradient descent optimization for the score model $D_\phi$ (optimizing Equation (13)) and the discriminator $C_\psi$ (optimizing Equation (15)) before one gradient descent optimization for the generator $D_\theta$. To stabilize training, we employ adversarial loss warmup by disabling adversarial loss at the beginning of training. The distribution of $\sigma$ for the noisy data condition $\mathbf{y}_\sigma$ is defined by sampling uniformly on EDM (Karras et al., 2022) inference time noise schedule, given by

$$\sigma_i = \left(\sigma_{\max}^\rho + \frac{i}{N-1}(\sigma_{\min}^\rho - \sigma_{\max}^\rho)\right)^{\frac{1}{\rho}}, \; i = 0, 2, ..., N-1, \tag{42}$$

where we select $\sigma_{\max} = 80.0$, $\sigma_{\min} = 0.002$, $\rho = 7.0$, and $N = 1000$. The distribution of $t$ is defined using LogNormal distribution (Karras et al., 2022) as $\log t \sim \mathcal{N}(P_{\mathrm{mean}}, P_{\mathrm{std}}^2)$ where $P_{\mathrm{mean}}, P_{\mathrm{std}}$ are hyperparameters. We provide detailed training hyperparameters in Table 3.

**Class-conditional generation** The class-conditional image generation on ImageNet datasets is achieved by performing denoising posterior sampling from pure Gaussian noise at a sufficiently high noise level $\sigma_{\mathrm{init}}$. We set $\sigma_{\mathrm{init}} = 80.0$ for all experiments. To specifying the noise schedule $\sigma_i$ for multi-step sampling proposed in Section 3.3, we select time steps $i$ and decide the noise level $\sigma_i$ using the EDM inference time noise schedule given in Equation (42) with $N = 40$, $\sigma_{\min} = 0.002$, $\sigma_{\max} = 80$, and $\rho = 7.0$. Detailed time steps for each experiment can be found in Table 3.

Table 4. Hyperparameter details for PnP-GD.

| | Inpaint (Box) | Deblur (Gaussian) | Deblur (Motion) | Super resolution | Phase retrieval |
|---|---|---|---|---|---|
| **Annealing schedule details** | | | | | |
| Steps ($N$) | 50 | 50 | 50 | 50 | 50 |
| $\sigma_{\max}$ | 80.0 | 80.0 | 80.0 | 80.0 | 80.0 |
| $\sigma_{\min}$ | 0.002 | 0.002 | 0.002 | 0.002 | 0.002 |
| $\rho$ | 2.0 | 2.0 | 2.0 | 2.0 | 2.0 |
| **EMA schedule details** | | | | | |
| EMA threshold ($\sigma_{\text{ema}}$) | $\infty$ | $\infty$ | $\infty$ | $\infty$ | 0.2 |
| EMA decay ($\mu$) | 0.2 | 0.6 | 0.6 | 0.6 | 0.6 |
| **Likelihood step details** | | | | | |
| Energy strength $\beta$ | 1e-4 | 2e-3 | 4e-3 | 1e-3 | 1e-3 |
| ULA step | - | - | - | 100 | 100 |
| ULA $C_1$ | - | - | - | 0.1 | 0.1 |
| ULA $C_2$ | - | - | - | 0.1 | 0.1 |

### B.3. Inverse Problem Solving

**Model Training** To train a generative denoiser on FFHQ dataset using NCVSD, we first pretrain a XS size diffusion model using EDM2 codebase. The hyperparameters setup mostly following XS size EDM2 model for ImageNet-64×64 dataset, except that we use batch size of 128 and learning rate warmup of 1M images. Additionally, we implement loss scaling of 0.1 to prevent fp16 overflows. We train the model until FID plateaus, which takes roughly 32M training images. Training generative denoiser on FFHQ dataset is the same as on ImageNet dataset. The training hyperparameters can be found in Table 3.

**Likelihood Step** The general model for inverse problems is given as follows:

$$\mathbf{y} = \mathcal{A}(\mathbf{x}_0) + \mathbf{n}, \ \mathbf{n} \sim \mathcal{N}(0, \sigma_{\mathbf{y}}^2 \mathbf{I}), \tag{43}$$

where $\mathcal{A}$ is the degradation operator, which is possibly nonlinear, and $\mathbf{n}$ is an additive white Gaussian noise with std of $\sigma_{\mathbf{y}}$. Using Bayesian framework for solving inverse problems by formulating the posterior distribution $q(\mathbf{x}_0) \propto q_{\text{data}}(\mathbf{x}_0) q(\mathbf{y}|\mathbf{x}_0)$, the likelihood function is given by Gaussian likelihood as $q(\mathbf{y}|\mathbf{x}_0) = \mathcal{N}(\mathbf{y}|\mathcal{A}(\mathbf{x}_0), \sigma_{\mathbf{y}}^2 \mathbf{I})$. With the energy function formulation used in PnP-GD (Equation (17)), it is equivalent to define the energy as

$$\mathcal{E}(\mathbf{x}_0) := \|\mathbf{y} - \mathcal{A}(\mathbf{x}_0)\|_2^2, \ \beta := 2\sigma_{\mathbf{y}}^2. \tag{44}$$

However, for better empirical performance, we also consider $\beta$ as a hyperparameter to tune, following DAPS (Zhang et al., 2024). To accelerate the likelihood step, we use fast closed-form solvers implemented in PnP-DM (Wu et al., 2024) for linear inverse problems. For nonlinear inverse problems, we use ULA with adaptive step size presented in Section 4.

**Hyperparameters** For the annealing schedule $\sigma_i$ in PnP-GD (Algorithm 1), we use EDM inference time noise schedule given in Equation (42). We provide the detailed hyperparameters of PnP-GD in Table 4.

### B.4. Baseline Details

**DDRM** We borrow the results reported in DAPS (Zhang et al., 2024).

**DPS** We borrow the results reported in DAPS (Zhang et al., 2024).

**DiffPIR** For noisy super-resolution, Gaussian deblurring, and motion deblurring, we use the default setting in the original paper. For noisy inpainting, we use $\lambda = 7.0$, $\zeta = 1.0$, and 100 NFE.

**ΠGDM** We use the ΠGDM implementation provided in the codebase of Peng et al. (2024) since the original code base does not support noisy linear inverse problems.

*Table 5.* SSIM comparisons on noisy inverse problems. The results are averaged over 100 images. We use **bold** and underline when the proposed method (PnP-GD) achieves the best and the second best, respectively.

| Method | Inpaint (box) | Deblur (Gaussian) | Deblur (motion) | Super resolution | Phase retrieval |
|---|---|---|---|---|---|
| DDRM (Kawar et al., 2022) | 0.801 | 0.732 | 0.512 | 0.782 | - |
| DPS (Chung et al., 2023) | 0.792 | 0.764 | 0.801 | 0.753 | 0.441 |
| ΠGDM (Song et al., 2023a) | 0.663 | 0.720 | 0.733 | 0.720 | - |
| DWT-Var (Peng et al., 2024) | 0.796 | 0.795 | 0.798 | 0.802 | - |
| DAPS (Zhang et al., 2024) | 0.814 | 0.817 | 0.847 | 0.818 | 0.851 |
| PnP-DM (Wu et al., 2024) | - | 0.780 | 0.795 | 0.787 | 0.628 |
| PnP-GD (*Proposed*) | **0.814** | 0.777 | 0.801 | 0.805 | 0.797 |

**DWT-Var**   We report the results based on the original codebase of Peng et al. (2024)[4] with the default settings. For box inpainting, we use Type II guidance with the DWT-Var only used when the std of the diffusion noise is below $0.5$.

**DAPS**   We report the results based on the DAPS codebase[5] with the default settings.

**PnP-DM**   We compare with PnP-DM using EDM as priors, i.e., PnP-DM (EDM) in the original paper. For linear inverse problems, we use the default setting and report the metrics based on single sample instead of the mean over 20 samples for a more direct comparison to PnP-GD. For phase retrieval, we use $\sigma_{\mathbf{y}} = 0.05$ instead of $\sigma_{\mathbf{y}} = 0.01$ of the original paper for fair comparisons.

# C. Additional Results

**Training FID curves**   In Figure 3, we plot the training FID v.s the number of training images, demonstrating classic training time scaling law and the effectiveness of test time scaling.

**Qualitative samples for class-conditional image generation on ImageNet 512×512**   In Figures 4-5, we show qualitative results of 4-step samples generated by NCVSD-L for class-conditional image generation on ImageNet-512×512 dataset.

**SSIM comparisons for inverse problem solving**   In Table 5, we provide additional SSIM comparisons of different methods.

**Effectiveness of EMA rate in PnP-GD**   In Figure 6, we plot the LPIPS and PSNR values under different EMA decay rates $\mu$ used in PnP-GD. As shown, PSNR tends to favor larger values of $\mu$, as they give more weight to historical samples, making the final results closer to the *posterior mean*. This approach tends to produce blurrier images but with less distortion. In contrast, LPIPS favors smaller values of $\mu$, making the final results closer to the *posterior samples*, which, while reducing blur, can lead to higher distortion. For visualization, please refer to Figure 7.

**Qualitative samples for inverse problem solving**   In Figure 8, we present visual comparisons with different methods for inverse problem solving. As can be seen, our method (PnP-GD) generate samples with more high frequency details compared to baselines.

---

[4]https://github.com/xypeng9903/k-diffusion-inverse-problems
[5]https://github.com/zhangbingliang2019/DAPS

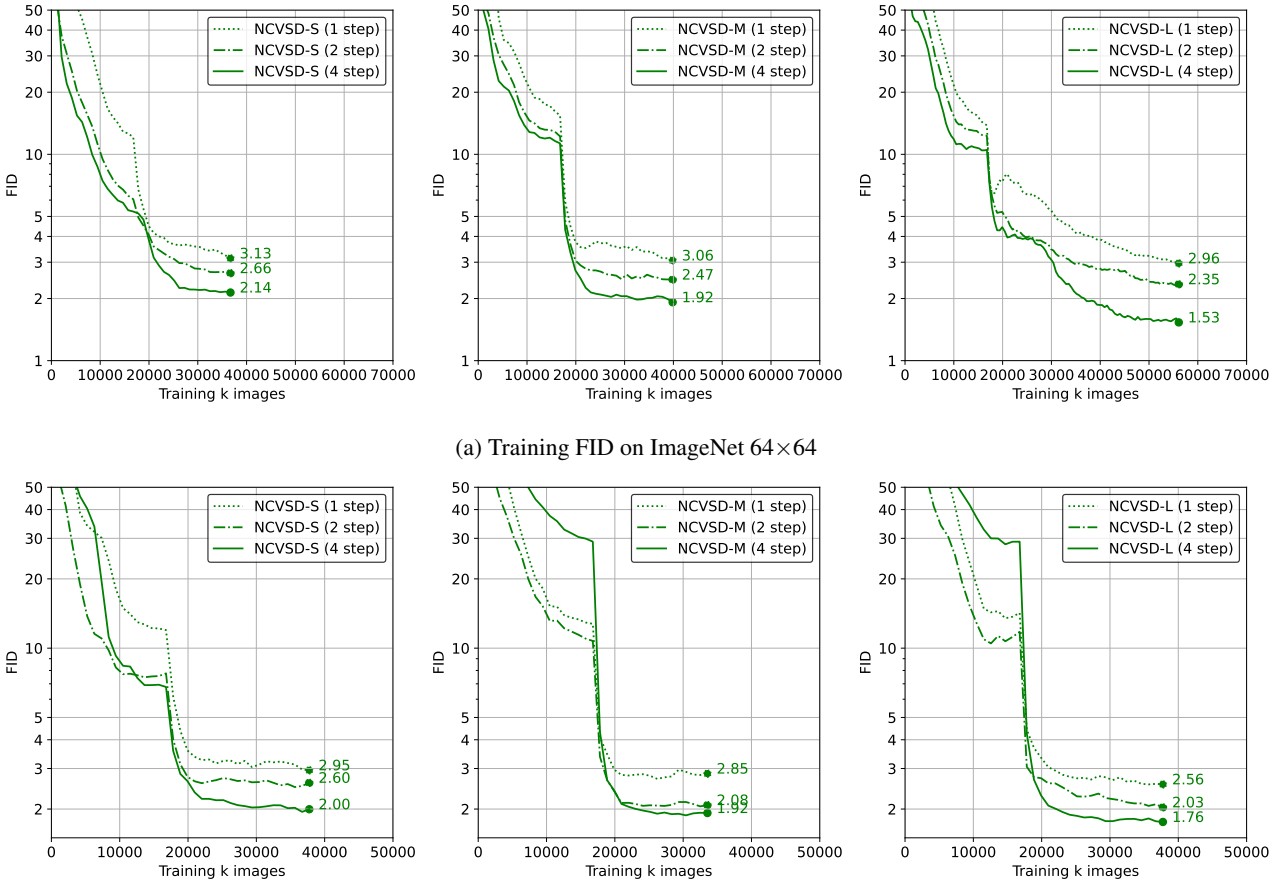

(a) Training FID on ImageNet 64×64

(b) Training FID on ImageNet 512×512

Figure 3. FID v.s the number of training images.

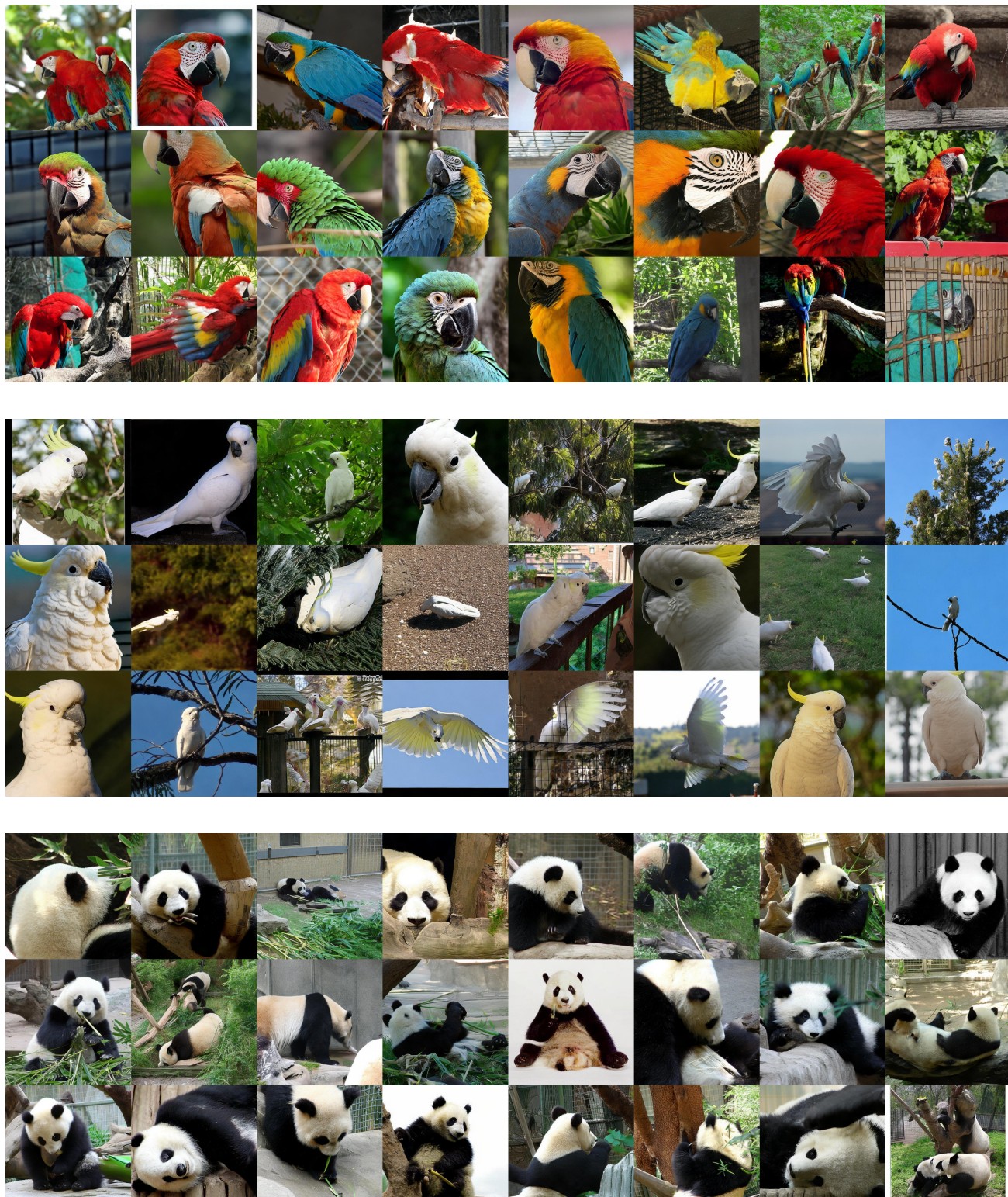

*Figure 4.* Uncurated 4-step samples generated by NCVSD-L for class-conditional ImageNet $512 \times 512$ generation. Top: class 88 (macaw); middle: class 89 (cokatoo); bottom: class 388 (giant panda).

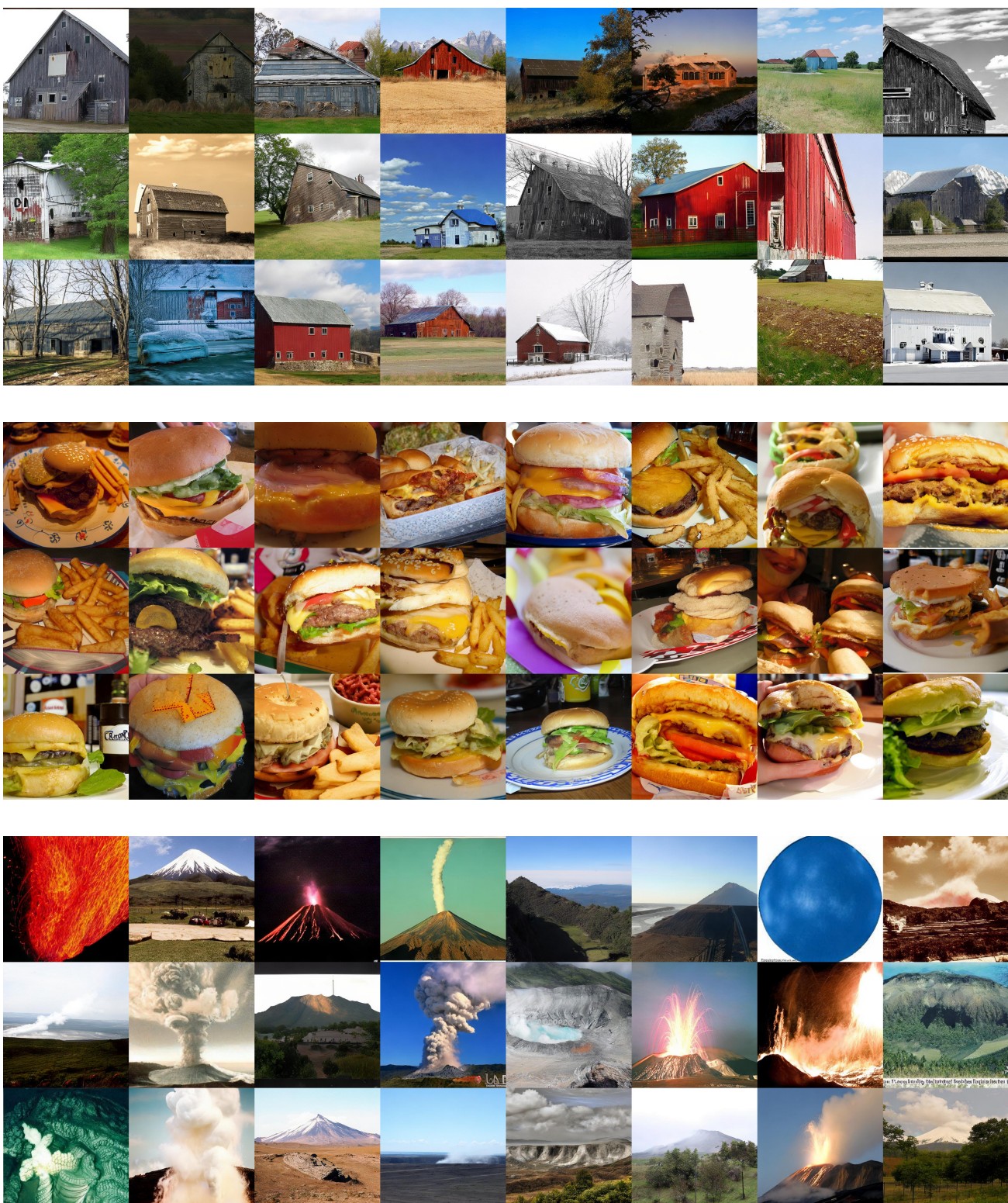

*Figure 5.* Uncurated 4-step samples generated by NCVSD-L for class-conditional ImageNet 512×512 generation. Top: class 425 (barn); middle: class 933 (cheeseburger); bottom: class 980 (volcano).

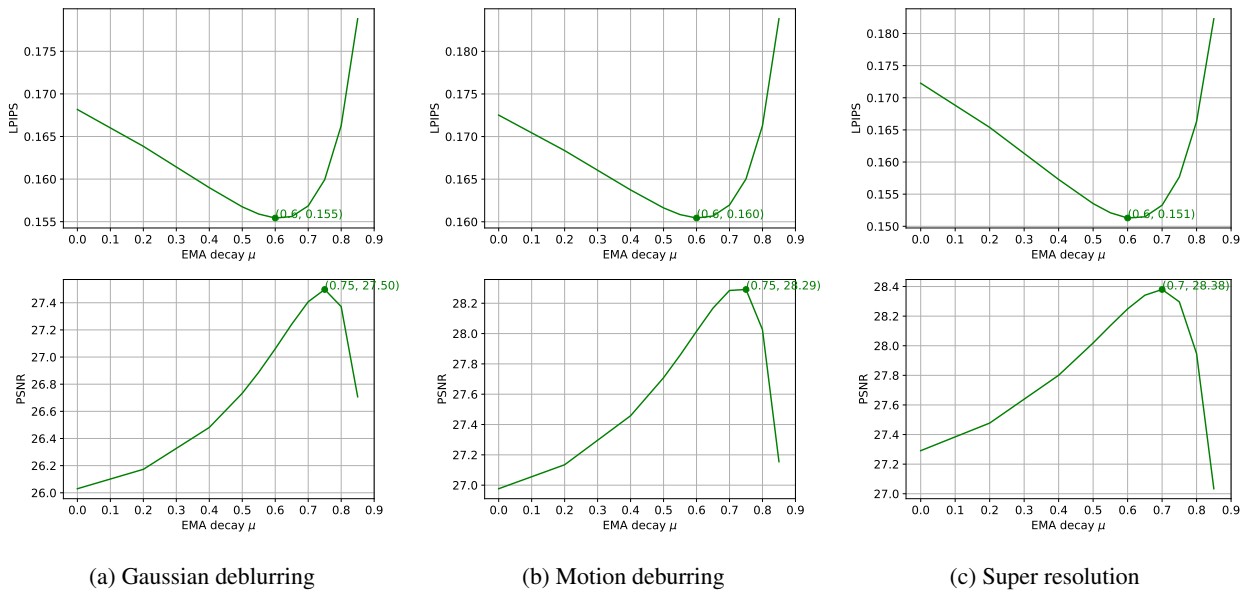

(a) Gaussian deblurring

(b) Motion deburring

(c) Super resolution

*Figure 6.* Effectiveness of EMA rate in PnP-GD.

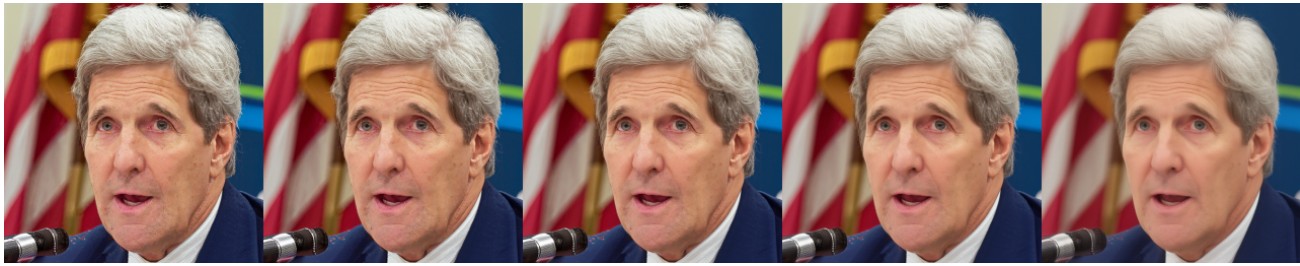

*Figure 7.* Samples of PnP-GD under different EMA decay rate $\mu$. From left to right, $\mu$ equals to 0.0, 0.2, 0.4, 0.6, 0.8.

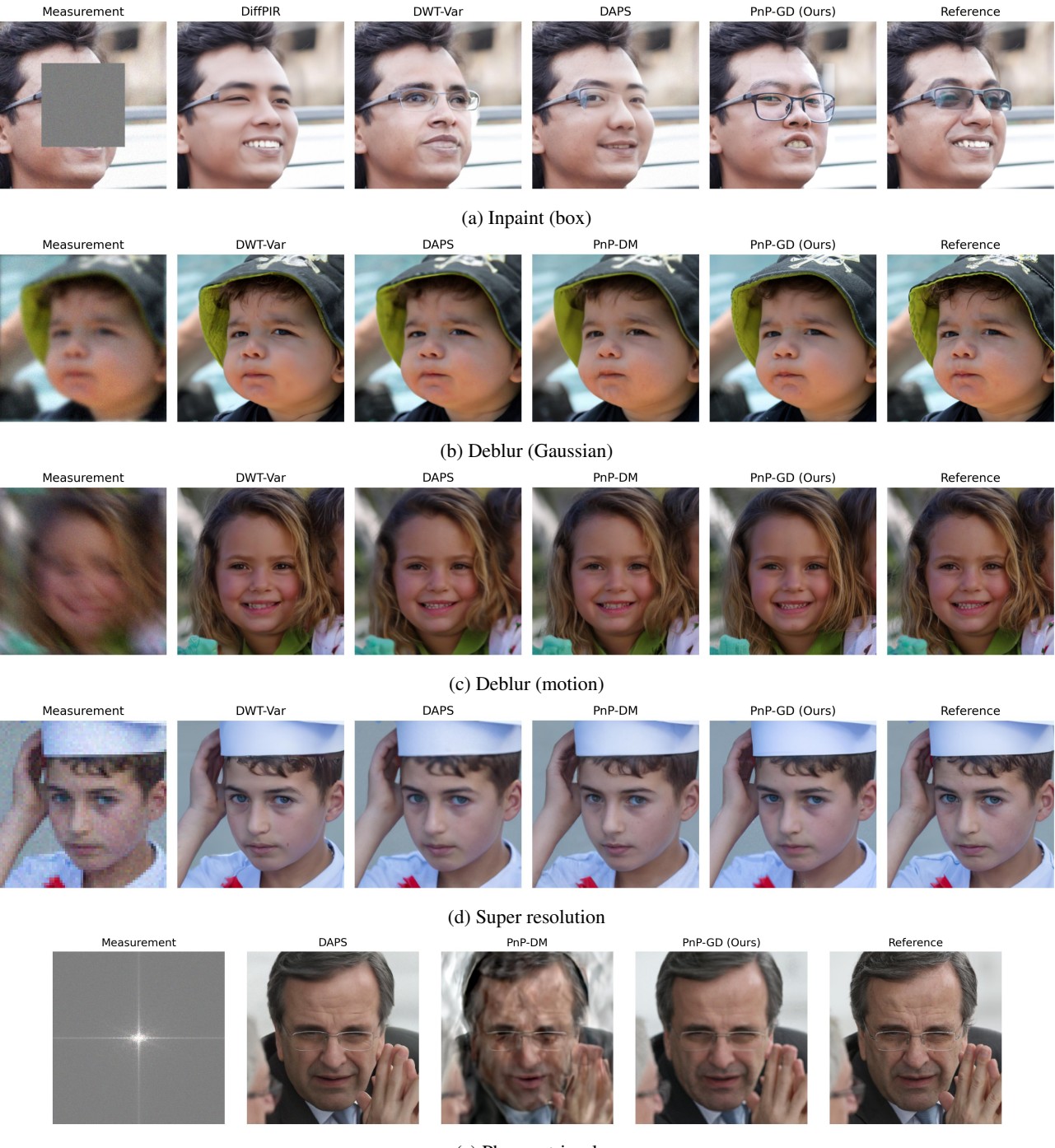

*Figure 8.* Visual comparisons for inverse problem solving.

