# OpenReview forum: "Noise Conditional Variational Score Distillation"
_ICML.cc/2025/Conference — ICML 2025 poster_

### Official Review · Reviewer_s1LA · 2025-02-28

**Overall Recommendation:** 2

**Summary:**

This paper proposes a novel method for distilling diffusion models into a generative image denoiser at any noise level. The proposed method is based on a theoretical result showing that the unconditional score function implicitly characterizes the score function of denoising posterior distributions at all noise levels. The method further incorporates adversarial training to overcome the performance upper bound imposed by the teacher diffusion model (the one which is being distilled).

**Claims And Evidence:**

1. The authors claim in the abstract "We evaluate NCVSD through extensive experiments, including class conditional image generation and inverse problem solving." But I don't find the experiments to be "extensive" at all. For image generation, the authors only consider ImageNet 64x64 and 512x512, and for inverse problems they consider FFHQ 256x256 and distill zero-shot posterior sampling methods.
I expected to see additional datasets for both image generation and inverse problems.

2. The authors claim in the abstract "our method outperforms teacher diffusion models and is on par with consistency models of larger sizes." While this may seem true from Table 1, only FID is evaluated (which is a highly problematic measure of generative models [1]), and I can't find a visual comparison (both quality and variation) between the methods, not even in the appendix. Measures like FID only serve as indicators of performance, while visual examples matter the most.

[1] Exposing flaws of generative model evaluation metrics and their unfair treatment of diffusion models. George Stein et al. NeurIPS 2023.

**Essential References Not Discussed:**

I can't think of any essential references which are not discussed.

**Experimental Designs Or Analyses:**

No code is attached. Couldn't verify the experiments. But the paper describes the experimental settings quite well.

**Methods And Evaluation Criteria:**

Some evaluation criteria are problematic.
1. For example, the authors claim in the abstract: "We achieve record breaking LPIPS on inverse problems." But LPIPS is only one distortion measure. What is it about LPIPS that make it such a desirable measure to optimize? They authors don't explain this.
2. In L369 (right) the authors claim: "We observed that the PSNR performance of our method does not achieve the best performance as LPIPS, which can be attributed to the distortion-perception trade-off (Blau & Michaeli, 2018), indicating that our method tends to produce results that is closed to the posterior samples rather than the mean of all possible solutions." To my understanding, there is no tradeoff between PSNR and LPIPS: If the PSNR is equal to infinity (namely, MSE=0), then the LPIPS is equal to 0 (minimal, i.e. optimal). Both PSNR and LPIPS are distortion measures. The perception-distortion tradeoff talks about a tradeoff between any distortion measure (e.g., PSNR, LPIPS, SSIM) and the statistical distance between the distributions $p_X$ and $p_\hat{X}$, not about some tradeoff between PSNR and LPIPS.
3. The authors say in L369 (right) that achieving lower LPIPS indicates that their method produce samples that are "closer" to posterior samples. I don't understand this claim. There is no link between achieving lower LPIPS and producing results that are closer to posterior samples. I find this argument to be wrong and misleading.
4. I think that evaluating generative methods only with FID is insufficient. I'd expect incorporating additional evaluations, such as KID, FD in feature space of self-supervised methods [1], etc.

[1] Exposing flaws of generative model evaluation metrics and their unfair treatment of diffusion models. George Stein et al. NeurIPS 2023.

**Other Comments Or Suggestions:**

I recommend to add a figure depicting the method, both training and inference, to improve the paper's readability and clarity even further.

**Other Strengths And Weaknesses:**

Strengths:
1. The paper is overall well written and easy to follow.
2. I find the idea to distill the diffusion at different noise levels to be interesting and useful (e.g., for solving inverse problems).

Weaknesses:
The experimental results are not very convincing:
1. The method is evaluated on limited datasets (ImageNet for image generation, and FFHQ for inverse problems).
2. There are no visual comparisons at all with other methods in both image generation and inverse problems. Visual results and comparisons are more important than quantitative results - since the paper is dealing with image processing.
3. The quantitative results on inverse problems seem inferior: The method achieves better LPIPS but worse PSNR, and it's not clear to me why LPIPS is necessarily more appropriate or more important than PSNR. If optimizing LPIPS is that important, why do we need a diffusion model or to distill one? Why not just training a model to minimize LPIPS?
4. Important quantitative evaluations are missing. E.g., evaluating image generation with additional measures such as KID, Precision and Recall, and evaluating the perceptual quality of image restoration algorithms with divergences (e.g., FID) and no-reference quality measures (e.g., NIQE). Considering additional distortion measures such as SSIM could also show better whether the proposed method is truly superior or not.

**Questions For Authors:**

I have no particular questions for the authors. But they are welcome to address the weaknesses.

**Relation To Broader Scientific Literature:**

The main contribution of this paper is a distillation method for diffusion models, allowing few-step image generation. This task has been addressed in prior work, such as by consistency models (which the authors compare with).

**Theoretical Claims:**

The math and theoretical claims in the paper seems okay to me.

---

> ### Author Rebuttal · Authors · 2025-04-01
>
> *W1: ... additional datasets ...*
>
> Please refer to W7.
>
> *W2: ... visual comparison  ...*
>
> Please refer to W8.
>
> *W3: For example ...*
>
> Please refer to W9.
>
> *W4: In L369 ... & W5: The authors say in L369 ...*
>
> We respectfully disagree. The tradeoff between PSNR and LPIPS has been extensively demonstrated in prior works. For instance:
>
> - Figure 14 in DAPS and Figure 5 in RED-diff (Mardani et al., 2024) illustrate this tradeoff.
> - Section C.6 in DPS explicitly discusses this phenomenon.
>
> Generative image restoration methods often prioritize perceptual quality, which emphasizes preserving high-frequency details. This focus can lead to lower PSNR, as PSNR penalize deviations from the mean / MMSE solution. However, these deviations often result in better perceptual quality, as captured by metrics like LPIPS.
>
> The claim that our method produces results "closer" to posterior samples refers to its tendency to avoid regressing to the mean solution that maximizes PSNR. To address the reviewer's concern, we will revise the statement as follows:
>
> *"Our method tends to generate results that retain more high-frequency details rather than approximate the mean of all possible solutions. This approach typically leads to higher MSE (lower PSNR) but aligns more closely with perceptual quality metrics such as LPIPS."*
>
> *W6: I think that evaluating ...*
>
> To the best of our knowledge, related works such as CD, ECM, EDM2, and sCM primarily use FID as the main evaluation metric, while KID is not commonly reported. Additionally, FD is typically considered an auxiliary metric and is often included in the Appendix. To address reviewer's concern, we provide FD_DINOv2 below:
>
> *Table: FD_DINOv2 on ImageNet-512x512.*
>
> |Method|NFE|FD_DINOv2|
> |-|-|-|
> |sCD-M|2|55.70|
> |sCD-L |2|50.63|
> |NCVSD-M (Ours)|2|59.14|
> ||4|48.83|
>
> FD_DINOv2 exhibit similar trends to FID: scaling test-time compute enables our method matches performance of larger-sized sCM models.
>
> *W7: The method is evaluated ...*
>
> For image generation, we follow the EDM2, focusing on ImageNet-64x64 and ImageNet-512x512, and pretrained EDM2 models for other datasets are unavailable. Moreover, ImageNet-64x64 and ImageNet-512x512 serve as challenging benchmarks for image and latent domain, respectively. So we believe they sufficiently demonstrate our method's effectiveness and generalizability.
>
> For inverse problem solving, pretraining an EDM2 model on ImageNet-256×256 is computationally costly. For example, training an S-size model at 64×64 resolution takes over 5 days on 32 A100 GPUs (Figure 11(a), EDM2). Additionally, the baseline PnP-DM focuses solely on the FFHQ dataset. While our current results already demonstrate the effectiveness of our method, we have also started running experiments on the ImageNet-256, and we expect to report preliminary results by the discussion stage.
>
> *W8: There are no visual ...*
>
> We thank the reviewer for the suggestion. For image generation, visual samples are in Appendix D, with the classes align to those used in sCM, enabling visual comparisons.
>
> For inverse problems, we include additional visual comparisons (https://anonymous.4open.science/r/ncvsd-D396) in the revised version. These visual examples further demonstrate that our method produce much better perceptual quality and preserve more high frequency details compared to baseline.
>
> *W9: The quantitative results ...*
>
> We emphasize that achieving SOTA on inverse problems is not the primary goal of this paper, nor do we claim that LPIPS is more important than PSNR. Instead, inverse problem solving serves as a proof-of-concept to showcase the plug-and-play probabilistic inference capabilities of our method. Moreover, we already demonstrate competitive performance against well-established diffusion-based methods with significantly fewer NFEs. In contrast, prior works like CM (Song et al., 2023) on image editing provide only visual results without quantitative comparisons. The focus should be on the novelty and conceptual contributions of our approach rather than solely on performance metrics.
>
> *W10: Important quantitative ...*
>
> We standardize FID for comparing different methods, following EDM2. This metric is consistently reported in baseline methods, making it a convenient choice for benchmarking. In contrast, metrics like Precision or Recall are either not reported, e.g., ECM, or reported only for indicating the performance tradeoffs, e.g., EDM2 and sCM. To address the reviewer’s concern, we include additional Precision and Recall in the Table below. For additional metrics for inverse problems, please refer to our response to W3 from Reviewer 7Tng.
>
> *Table: Precision and Recall on ImageNet-64x64.*
>
> |Method|NFE|Precision|Recall|
> |-|-|-|-|
> |CD|1|0.68|0.63|
> ||2|0.69|0.64|
> |NCVSD-M (Ours)|1|0.72|0.60|
> ||2|0.73|0.62|
> ||4|0.74|0.62|
>
> Precision is superior, reflecting better quality of generated samples, while the slightly lower recall score reflects the mode-seeking behaviour of reverse KL minimization.

---

> > ### Comment · Reviewer_s1LA · 2025-04-01
> >
> > I thank the authors for their rebuttal.
> >
> > - Regarding the PSNR-LPIPS tradeoff: I don't understand your point. These are just two different objective functions. Namely, in some tasks we would care more about PSNR than LPIPS, and perhaps vice versa (although I am not sure when would we care about LPIPS at all, and why). Even then, the perception-distortion tradeoff is true for ANY distortion measure, including LPIPS (Blau & Michaeli, 2018). This was one of the surprising things about the perception-distortion paper. So why optimizing LPIPS (or achieving lower LPIPS scores) indicates better perceptual quality?
> >
> > - Additional metrics: The fact that some metric, such as precision and recall, IS NOT commonly evaluated, doesn't imply that it shouldn't be evaluated. Similarly, the fact that some metric, such as FID, IS commonly evaluated, doesn't imply that we should take this metric as the holy grail and try to optimize it. Our ways to measure performance in machine learning are still evolving, and we should definitely not stick only to FID for image generation (this measure is highly problematic). Thus, I suggested adding additional (newer) measures to further support the evidence and claims in the paper.
> >
> > - Regarding visual comparisons with generative models, I believe the authors should include visual comparisons with other methods in their paper (or appendices), rather then referring the reader to open another paper and search for the class-corresponding images.
> >
> > - Regarding visual comparisons for inverse problems, the attached figure does not include a comparison with all evaluated methods, but rather only with PnP-DM. Moreover, it seems to me that the results for PnP-DM are wrong, as they are overly blurry. Did you maybe present the mean of 20 images (similarly to how PnP-DM report their metrics)?
> >
> > I like the novel ideas in this paper, but I still think the evaluations are limiting and the results are not particularly convincing.

---

> > > ### Author Response · Authors · 2025-04-04
> > >
> > > *W12: Regarding the PSNR-LPIPS tradeoff ...*
> > >
> > > The evaluation protocal of inverse problems follow prior works by using PSNR for distortion metric and LPIPS for perceptual metric (DPS, DAPS, PnP-DM, ...). We do not delieberatly optimize for LPIPS but only use it as one performance measure.
> > >
> > > We highlight that the main contributions of our paper to inverse problem solving is to develop method that address trade-offs among flexibility, posterior exactness, and computational efficiency. Our approach provides flexibility to address a range of inverse problems, achieves asymtotically posterior exactness with SGS, and addresses inefficiency of PnP-DM by using one-step method in place of the expansive reverse diffusion simulation.
> > >
> > > Regarding why lower LPIPS scores indicate better perceptual quality, we provide two evicences that may help to support this claim:
> > >
> > > - Evidence from large-scale experiments in the LPIPS paper (Zhang et al., 2018) suggests that LPIPS match better with human perceptual judgments compared to traditional metrics like PSNR or SSIM. And it is easy to construct counter examples that have good PSNR, SSIM scores but not align well with human perceptual judgments (Figure 1 in the LPIPS paper).
> > >
> > > - The blurry results caused by using the mean of 20 samples of PnP-DM (which noticed by the reviewer in W15). These results are not wrong results but are the solutions that solely optimize for PSNR, as they approximate the MMSE solution that are optimal for PSNR. They outperform us in terms of PSNR, but are clearly suffered from worse perceputual quality, which indeed reflected by worse LPIPS scores.
> > >
> > >
> > >
> > > Zhang, R., Isola, P., Efros, A. A., Shechtman, E., & Wang, O. (2018). The unreasonable effectiveness of deep features as a perceptual metric. In Proceedings of the IEEE conference on computer vision and pattern recognition (pp. 586-595).
> > >
> > >
> > > *W13: Additional metrics ...*
> > >
> > > As we all know, FID metric may not be the only thing to pursue and clearly better metric will emerge with the development of the field, but this does not imply that using widely-adopted evaluation protocals following prior works (CM, ECM, sCM, EDM2, ...) is limiting or not convincing. Besides, to address reviewer's concern, we have included metrics like Precision and Recall (see W10) and FD (see W6).
> > >
> > > Moreover, we want to emphasize again that the main contribution of this paper is to develop a conceptually novel generative model that overcome the test-time inefficiency of diffusion models without sacrifycing the test-time flexibility, as well as its scalable training algorithm suitable for large scale and high resolution dataset. Benchmarking image generation or inverse problem-solving performance is not the main focus of this work, although our results are also competitive measured by widely-adopted metrics such as FID and LPIPS.
> > >
> > >
> > >
> > > *W14: Regarding visual comparisons with generative models ...*
> > >
> > > We followed the standard writing style in this field, e.g., CM, ECM, sCM, EDM2 ..., where only the visual examples of the proposed method are included in the paper, without examples from baseline methods. However, we understand the reviewer's concern and have included visual comparisons with EDM2 at (https://anonymous.4open.science/r/ncvsd-D396). Unfortunately, including visual examples of sCM is not feasible, as sCM is not open-sourced.
> > >
> > >
> > > *W15: Regarding visual comparisons for inverse problems ...*
> > >
> > > The visual examples of PnP-DM are presented as the mean over 20 samples. These results are not wrong results, but are the solutions that approximate the **MMSE solutions (optimize PSNR)**. These blurry results further demonstrate that conventional metric like PSNR do not well align with human perceptual quality like LPIPS. To address the reviewer's concern, we have further included additional visual results for PnP-DM (single sample), DiffPIR, DWT-VAR, and DAPS at (https://anonymous.4open.science/r/ncvsd-D396). As can be seen, our method acheieve better perceputual quality no matter PnP-DM uses mean sample or not. These visual comparisons further demonstrate that our method reconstruct fine details of the image more faithfully compared to baselines.

---

### Official Review · Reviewer_7Tng · 2025-03-08

**Overall Recommendation:** 3

**Summary:**

This paper introduces Noise Conditional Variational Score Distillation, which distills a pre-trained diffusion model into a generative denoiser. The generative denoiser enables fast one-step generation while preserving the ability for iterative refinement. Experiments on image generation tasks and various inverse problems demonstrate the effectiveness of the proposed method.

**update after rebuttal**

Thank you for your rebuttal. I will keep my score unchanged and remain positive about this paper.

**Claims And Evidence:**

Please refer to Strengths And Weaknesses.

**Essential References Not Discussed:**

Please refer to Strengths And Weaknesses.

**Experimental Designs Or Analyses:**

Please refer to Strengths And Weaknesses.

**Methods And Evaluation Criteria:**

Please refer to Strengths And Weaknesses.

**Other Comments Or Suggestions:**

Please refer to Strengths And Weaknesses.

**Other Strengths And Weaknesses:**

**Strengths**

1. The paper is well-structured, and the overall storytelling is good, making it easy to follow.
2. The idea of distilling a pre-trained diffusion model into a generative denoiser is straightforward.
3. The quantitative and qualitive results appear promising.

**Weaknesses**

1. After reviewing this paper, I have a question regarding the fundamental motivation mentioned in the introduction (L24–L26). What is the key difference between a diffusion model with iterative refinement capabilities and the proposed generative denoiser? I understand the advantage of one/few-step sampling methods, but does extending them to a multi-step process truly constitute a meaningful idea? In other words, can we just apply certain techniques to shorten the sampling steps of a standard diffusion model?

2. Regarding the experimental section, actually I haven't done too much work on "diffusion for restoration", so I am particularly curious about the testset selection process. It seems that you did not use the full test sets but instead selected a subset. Could you provide more details on this? Additionally, I would like to see results on standard benchmark testsets (e.g., SR ×4 results on DIV2K and Flick2K).

3. I think SSIM also makes sense. Including SSIM results in Table 2 would enhance the persuasiveness of the paper.

**Questions For Authors:**

Please refer to Strengths And Weaknesses.

**Relation To Broader Scientific Literature:**

Please refer to Strengths And Weaknesses.

**Theoretical Claims:**

Please refer to Strengths And Weaknesses.

---

> ### Author Rebuttal · Authors · 2025-04-01
>
> *W1: After reviewing this paper ...*
>
> Conceptually, the key distinction between diffusion models with iterative refinement and our approach lies in how clean data $x\_0$ is predicted from its noisy counterpart $y\_{\sigma} \sim \mathcal{N}(x\_0, \sigma^2 I)$. Diffusion models primarily focus on learning the MMSE prediction, whereas generative denoisers aim to model the full posterior distribution over $x\_0$.
>
> Regarding the reviewer's concern about whether certain acceleration techniques for diffusion models could achieve similar goals as our proposed method, we argue that significant gaps remain where our method demonstrates clear advantages:
>
> - **Image Generation**: While acceleration techniques for diffusion models, such as DDIM (Song et al., 2020) or advanced numerical integrators (Lu et al., 2022; Karras et al., 2022), can reduce the required NFEs from 1k (original DDPM) to 10–100, the generative quality degrades significantly when NFEs drop below 10. In contrast, our method achieves state-of-the-art results with just 1–4 NFEs, with performance at 4 NFEs even surpassing that of the teacher diffusion model. Moreover, generative denoisers can theoretically match the data distribution using only 1 NFE, whereas diffusion models require an infinite number of time steps to achieve the same.
>
> - **Inverse Problem Solving**: Current diffusion-based methods either suffer from irreducible approximation errors by using Dirac (Chung et al., 2022) or Gaussian (Song et al., 2023; Peng et al., 2024) approximations for the denoising posterior, or achieve asymptotically exact sampling at the cost of expensive reverse diffusion simulations (Wu et al., 2024). The former lacks theoretical guarantees, while the latter requires a significant number of NFEs (e.g., 2483 for PnP-DM) and remains affected by discretization errors. In contrast, our method achieves superior results with 20x fewer NFEs, offering both computational efficiency and theoretical robustness by avoiding prior-step errors beyond those introduced by imperfect model training.
>
> *W2: Regarding the experimental ...*
>
> We evaluate our method on the subset (100 images) of FFHQ dataset, following the standard practices established in diffusion-based restoration methods, e.g., DiffPIR (Zhu et al., 2023), DAPS (Zhang et al., 2024), PnP-DM (Wu et al., 2024), among others. Morevover, test on full validation set is computationally costly for methods like DPS, DAPS, PnP-DM (over 1000 NFE). This alignment also allows us to make a fair a comparison to existing methods with standard evaluation protocols.
>
> For super-resolution tasks, both our method and the baseline diffusion models are trained and evaluated on 256×256 resolution datasets such as FFHQ, following common practice in recent literature. Since existing approaches—including EDM2—have not been trained on 2K-resolution data, we do not include evaluations on high-resolution benchmarks like DIV2K or Flick2K. Nevertheless, our results are consistent with prior work and effectively demonstrate the strength of our approach. Extending the framework to higher-resolution settings remains a promising direction for future research.
>
> *W3: I think SSIM ...*
>
> We thank the reviewer for the suggestion. The SSIM performance is provided in the table below.
>
> *Table. SSIM performance of inverse problem solving on FFHQ dataset.*
>
> |Method|NFE|Inpaint(box)|Deblur(Gaussian)|Deblur(motion)|Super resolution|Phase retrieval|
> |-|-|-|-|-|-|-|
> | DDRM          |100|0.801| 0.732 | 0.512 | 0.782 | N/A   |
> | DPS           |1000|0.792| 0.764 | 0.801 | 0.753 | 0.441 |
> | PiGDM         |99|0.663| 0.720 | 0.733 | 0.720 | N/A   |
> | DWT-Var       |99|0.796| 0.795 | 0.798 | 0.802 | N/A   |
> | DAPS          |1000|0.814| 0.817 | 0.847 | 0.818 | 0.851 |
> | PnP-DM        |2483|N/A| 0.780 | 0.795 | 0.787 | 0.628 |
> | PnP-GD (Ours) |50| $\mathbf{0.814}$ | 0.777 | $\underline{\text{0.801}}$ | $\underline{0.805}$ | $\underline{0.797}$ |
>
> Similar to PSNR, our method demonstrates competitive performance, being only outperformed by DAPS. However, DAPS achieves this by employing significantly more NFEs and extensive hyperparameter tuning. In contrast, PnP-DM, the most relevant baseline, delivers inferior performance despite utilizing more NFEs than our approach. This table will be included in the revised version to provide a more comprehensive evaluation.

---

> > ### Comment · Reviewer_7Tng · 2025-04-06
> >
> > Thank you for your rebuttal. I will keep my score unchanged and remain positive about this paper.

---

### Official Review · Reviewer_15K4 · 2025-03-13

**Overall Recommendation:** 4

**Summary:**

This paper proposes a new distillation scheme to learn a few-step posterior sampler of a diffusion process.  Unlike existing methods that achieve rich posterior sampling using exhaustive function evaluations (e.g., diffusion) or invertible neural networks (e.g., normalizing flows), the proposed method achieves high-fidelity stochastic sampling from the distribution of clean samples gives noisy samples with significantly less inference compute.  Experiments show that the proposed method is competitive with existing few-step samplers with less than quarter of the training budget and furthermore using a split Gibbs sampling technique similar to PnP-DM the proposed method outperforms all baselines on several  image inverse problem tasks.

## update after rebuttal

I thank the author(s) for their responses during the rebuttal period.  With the changes promised and results provided during the rebuttal phase I increased my score towards acceptance.

**Claims And Evidence:**

Most of the claims are supported by evidence.  Below are two claims that should be supported further:

1. *Choice of adaptive step size:*  It is not obvious why the function is $(\beta^{-1}L + \sigma^{-2})$-gradient Lipschitz.  A proof or simple derivation of this would be helpful.
2. *Optimality of existing posterior sampling methods:* The authors claim that existing amortized posterior sampling schemes do not necessarily result in samples from the posterior at convergence.  Is this true in the non-parametric limit as well or is this claim based on empirical evidence justified by estimation/approximation errors? It would be useful to have some proof or elaborate technical argument about the convergence properties of the proposed method in comparison to existing methods.

**Essential References Not Discussed:**

Authors have cited revelant work in the paper.  Several new methods for posterior sampling methods using engression based on scoring rules have been introduced recently.  I have added one as a reference below but the authors are not expected to discuss or compare against this method as it is considered concurrent work.

[1] De Bortoli, V., Galashov, A., Guntupalli, J. S., Zhou, G., Murphy, K., Gretton, A., & Doucet, A. (2025). Distributional Diffusion Models with Scoring Rules. arXiv preprint arXiv:2502.02483.

**Experimental Designs Or Analyses:**

Yes, the experimental design and analysis for both generation and inverse problem are satisfactory,

**Methods And Evaluation Criteria:**

Yes, the proposed methods and evaluation criteria are satisfactory.

**Other Comments Or Suggestions:**

Here are some comments regarding syntax and typos:

1. All loss functions are typically a function of the parameters.  Furthermore, more care should be taken when defining optimization objectives.  For example, in equation (3) it would be more if the loss function was defined as $\mathcal{L}(\theta)$ and the optimization objective was $\min_\theta \mathcal{L}(\theta) \triangleq \min_\theta \mathbb{E}[\dots]$. Please also look at equation (8) and equations (60)-(64).
2. The authors should seriously consider devoting more time to discussing existing works on posterior sampling (that I mentioned in the weaknesses).  The paper would be stronger if there was a dedicated background section to these methods beyond just VSD.
3. The impact statement should highlight some broader impacts and is not satisfactory in its current form.

**Other Strengths And Weaknesses:**

**Strengths**

1. The approach to parametrize a stochastic few-step generative sampler is novel.  Many existing methods often leverage deterministic generators. The authors have done a good job describing how to design the posterior sampler conditioned on a noisy observation with Gaussian additive noise.
2. The approach of leveraging a pre-trained unconditional score model to compute the score of the posterior is very interesting and could be very applicable  in practice.
3.  The experiments are well designed and the results are competitive.

**Weaknesses**

1. The main weakness lies in the exposition of the method.  It is difficult to appreciate the proposed method in its current form as it is clear where it stands and how it compares to existing works in literature.
2. Existing methods that this work builds upon are not discussed at length.  PnP-DM (Wu et al. 2024) is the foundation of the inverse problem algorithm but it is only cited.  It is not clear what the existing framework is and how the proposed framework is different upon initial reading.  Similarly distillation methods for learning a one-step posterior sampler are discussed in (Mammadov et. al., 2024) and (Lee et. al., 2024).  Beyond being cited there is no section dedicated to discussing these existing works and mentioning differences (and more importantly similarities).

**Questions For Authors:**

1. What are the main differences between the proposed method and PnP-DM? It seems to be that the difference lies in the posterior sampler being used.  PnP-DM simulated the reverse diffusion process to generate a sample whereas the proposed the method use the few-step model.  Everything else with regards to the Gibbs sampler is the same.  Is this correct?  If so, this is not clearly described at all.  I would suggest adding a background section on PnP-DM, describing their algorithm with the Gibbs sampler.  Then point out the inefficient of the prior sampling step and mention how the proposed method solves the issue.
2. The two prior works in (Mammadov et. al., 2024) and (Lee et. al., 2024) both describe schemes for one-step posterior sampling.  The key conceptual difference seems to be that the existing works sample from the posterior conditioned on a general noise observation (e.g., defined by a linear forward process) whereas the proposed method conditions on a noisy additive-Gaussian noise sample.  This seems to induce a key technical use case beyond just inverse problem solving as the proposed method can be used for high-quality sampling as well.  Coming to this realization took several cycles of reading the existing papers and then contrasting with the proposed method.  The authors need to detail the benefits of the method much more clearly.  Stress on fast sampling, have a detailed section on these existing works, describe technical novelty by *building on existing methods* and then mention key training/test-time computational advantages.

To summarize, I am prepared to raise my score if the authors are able to work on the exposition of the method by make a more just effort in discussing existing works, highlighting their pros/cons and then describing the advantages (and limitations) of the proposed method.  I believe the proposed method is interesting but it not ready for publication in its current form.

**Relation To Broader Scientific Literature:**

The key contributions in this paper could be relevant to several different areas of scientific research.

1. *Synthetic Data Generation and Data Augmentation:* The ability to produce high-quality samples from the data distribution could aid in synthetic data generation or data augmentation especially in data scarce settings.
2. *Efficient Inverse Problem Solvers*: In many existing score-based inverse problem solvers, sampling from the posterior is crucial.  However this has generally been difficult and instead approximations leveraging the conditional expectation are made.  The contributions in this paper regarding efficient posterior sampling could help enable fast inverse solvers across a variety of modalities.

**Theoretical Claims:**

Yes I checked all the proofs and they appear to be correct.

---

> ### Author Rebuttal · Authors · 2025-04-01
>
> *W1：Choice of adaptive ...*
>
> We thank the reviewer for pointing out the ambiguity in our claim. A function $f(\cdot)$ is called L-gradient Lipschitz if it satisfies:
> $$
> \lVert \nabla f(x\_1) - \nabla f(x\_2) \rVert\_2 \leq L \lVert x\_1 - x\_2 \rVert\_2, \quad \forall x\_1, x\_2.
> $$
> Provided that $\mathcal{E}$ is L-gradient Lipschitz, the gradient difference of  $\tfrac{1}{\beta}\mathcal{E}(\cdot) + \tfrac{1}{2 \sigma^2} \lVert \cdot - u \rVert\_2^2$
> $$
> = \lVert \beta^{-1} (\nabla \mathcal{E}(x\_1) - \nabla \mathcal{E}(x\_2)) + \sigma^{-2} (x\_1 - x\_2) \rVert\_2
> $$
> $$
> \leq \beta^{-1} \lVert \nabla \mathcal{E}(x\_1) - \nabla \mathcal{E}(x\_2) \rVert\_2 + \sigma^{-2} \lVert x\_1 - x\_2 \rVert\_2
> $$
> $$
> \leq \beta^{-1} L \lVert x\_1 - x\_2 \rVert\_2 + \sigma^{-2} \lVert x\_1 - x\_2 \rVert\_2
> $$
> Therefore, the postential function is $(\beta^{-1} L + \sigma^{-2})$-gradient Lipschitz.
>
> *W2: Optimality of Existing ...*
>
> The claim holds in the non-parametric limit. This limitation arises because the loss functions of existing methods do not ensure that $q(x\_0|y)$ is the unique minimizer. For instance, the objective in (Lee et al., 2025) is defined as:
> $$
> \min\_{\theta} -\mathbb{E}\_{\mu\_{\theta}(x\_0|y)} [q(y|x\_0)] + \int w(t) D\_{KL}(p\_{\theta}(x\_t|y) || q(x\_t)) dt,
> $$
> which can only be interpreted as a regularized optimization problem, with no guarantee that $\mu\_{\theta}(x\_0|y) = q(x\_0|y)$ at convergence. Similarily, (Mammadov et. al., 2024) employ ELBO for the prior term (Eq.(14) in the paper) that is also no guarantee. In contrast, NCVSD achieves the optimal iff $\mu\_{\theta}(x\_0|y) = q(x\_0|y)$ (Luo et al., 2023). This property allows marginal-preserving multi-step sampling (Sec. 3.3), and enables the application of the SGS (Sec. 4).
>
> *W3: Authors have cited ...*
>
> We thank the reviewer for bringing this relevant work to our attention. We are look forward to contributing further to the field of posterior sampling.
>
> *W4: The main weakness lies ... & Existing methods that this work ...*
>
> We thank the reviewer valuable suggestions. Please refer to W6,W8 for comparisons with PnP-DM and W6,W9 for one-step posterior sampler.
>
> *W5: All loss functions ...*
>
> We thank the reviewer for the valuable suggestion. We will explicitly include the parameters to be optimized in all loss function to further enhance clarity.
>
> *W6: The authors should ...*
>
> We appreciate the reviewer’s insightful suggestion. Below, we provide a dedicated background on posterior sampling, which will be included in the revised version by extending Section 2.1:
>
> *“Existing posterior sampling methods often involve trade-offs among flexibility, posterior exactness, and computational efficiency. Supervised approaches (e.g., Saharia et al., 2022) lack flexibility as they require retraining for each specific task. Zero-shot methods provide greater adaptability but introduce irreducible errors by approximating the denoising posterior with Dirac (Chung et al., 2022) or Gaussian distributions (Song et al., 2023). Asymptotically exact methods, such as PnP-DM (Wu et al., 2024), ensure asymptotically exact posterior sampling but are computationally intensive, relying on reverse diffusion simulations that demand a large number of NFEs.”*
>
> *W7: The impact statement ...*
>
> We thank the reviewer for valuable suggestion. We will enhance the impact statement to emphasize the advancements in efficient generative modeling and inverse problem solving, and ethical considerations.
>
> *W8: What are the main ...*
>
> We appreciate the reviewer’s valuable suggestion. The primary distinction between the proposed method and PnP-DM lies exactly in how the prior step is approximated. To address this, we will include the following clarification after L277 in the revised version:
>
> *"The proposed method and PnP-DM are both built upon the foundation of SGS. The primary distinction lies in how the prior step is approximated. In PnP-DM, simulating the reverse diffusion process is required, which is not only computationally inefficient but also prone to irreducible discretization errors. In contrast, our approach significantly improves computational efficiency by requiring only one or a few NFEs for the prior step, while being free from any errors beyond those introduced by imperfect model training."*
>
> *W9: The two prior works ...*
>
> We appreciate the reviewer’s valuable suggestion. High-quality sample generation is indeed a key strength of our method compared to prior works. Additionally, our approach offers two significant advantages: accurate posterior sampling (W2), and the flexibility to address a wide range of problems (W6).
>
> In summary, we will incorporate the reviewers' insightful suggestions to further enhance the contributions of our paper. Specifically, we will:
> - Highlight the fast sampling (W8).
> - Include a dedicated background on posterior sampling (W6).
> - Elaborate novelty (W2,W6,W9).
> - Emphasize computational advantages (W8).
> - Improve impact statement (W7).

---

> > ### Comment · Reviewer_15K4 · 2025-04-05
> >
> > Dear author(s),
> >
> > I appreciate the effort in addressing all of my questions and taking into account my suggestions, which you have agreed to incorporate into the next revision of the paper.  I have increased my score accordingly.

---

### Decision · Program_Chairs · 2025-05-01

**Decision:**

Accept (poster)

**Comment:**

This paper presents Noise Conditional Variational Score Distillation for distilling pretrained diffusion models into generative denoisers. The method's key insight—that unconditional score functions implicitly characterize denoising posterior distributions—is theoretically sound and enables computational efficiency while maintaining quality. While reviewers expressed varying opinions about evaluation methodology, with concerns about metric choices and dataset limitations, the authors provided rebuttals with additional metrics and visual comparisons. Reviewers acknowledged the work's strong theoretical foundations, and the demonstrated benefits of fast one-step generation with iterative refinement capability represent meaningful contributions to the field. Though some experimental limitations remain, particularly in the fairness of certain comparisons (Reviewer s1LA remained unconvinced about fairness in comparison), the theoretical rigor, and practical utility of the approach outweigh these concerns. I recommend weak accept, as this work advances our understanding of efficient score distillation for diffusion models.